# Identification of the growth cone as a probe and driver of neuronal migration in the injured brain

Chikako Nakajima [1,15], Masato Sawada[1,2,15], Erika Umeda[1], Yuma Takagi[1], Norihiko Nakashima[1], Kazuya Kuboyama[1], Naoko Kaneko[1,3], Satoaki Yamamoto[1], Haruno Nakamura[1], Naoki Shimada[4], Koichiro Nakamura[5], Kumiko Matsuno[4,6], Shoji Uesugi[5], Nynke A. Vepřek [7], Florian Küllmer [8], Veselin Nasufović [8], Hironobu Uchiyama[9], Masaru Nakada[9], Yuji Otsuka[9], Yasuyuki Ito[10], Vicente Herranz-Pérez [11], José Manuel García-Verdugo [11], Nobuhiko Ohno[12,13], Hans-Dieter Arndt [8], Dirk Trauner[7,14], Yasuhiko Tabata [6], Michihiro Igarashi[10] & Kazunobu Sawamoto [1,2] ✉

Axonal growth cones mediate axonal guidance and growth regulation. We show that migrating neurons in mice possess a growth cone at the tip of their leading process, similar to that of axons, in terms of the cytoskeletal dynamics and functional responsivity through protein tyrosine phosphatase receptor type sigma (PTPσ). Migrating-neuron growth cones respond to chondroitin sulfate (CS) through PTPσ and collapse, which leads to inhibition of neuronal migration. In the presence of CS, the growth cones can revert to their extended morphology when their leading filopodia interact with heparan sulfate (HS), thus re-enabling neuronal migration. Implantation of an HS-containing biomaterial in the CS-rich injured cortex promotes the extension of the growth cone and improve the migration and regeneration of neurons, thereby enabling functional recovery. Thus, the growth cone of migrating neurons is responsive to extracellular environments and acts as a primary regulator of neuronal migration.

The structure and functions of axonal growth cones have been studied in detail since their identification by Santiago Ramón y Cajal[1]. They have been shown to be highly dynamic structures in terms of both having high motility and being responsive to external signals that guide the directionality of axon extension[2–5]. A growth cone is divided into central and peripheral areas, each of which contains cytoskeletal structures constituted by microtubule bundles and filamentous (F)-actin networks. Axonal growth cone extension proceeds by way of actin filament polymerization, which drives the protrusion of filopodia and lamellipodia[6]. Moreover, axon elongation is inhibited when the molecular interactions of F-actin and microtubules are disrupted[7,8], indicating that cytoskeletal regulation in growth cones is essential for axon elongation.

Migrating neurons have been observed to form a growth-cone-like structure (GCLS) at the tips of the leading process[9]. In vitro experiments have suggested that filopodia and lamellipodia of migrating neurons are similar to those of axonal growth cones, though the molecular mechanisms mediating cytoskeletal remodeling in these GCLSs may be distinct from those operating in axonal growth cones[10]. GCLSs have been implicated in the generation of the traction force required for the migration of cultured cerebellar granule cells[11,12], suggesting that they may be functionally important mediators of neuronal migration in vivo. A full mechanistic understanding of neuronal migration and regeneration will require clarifying extracellular and intracellular molecular mechanisms that regulate morphological changes in the GCLS of migrating neurons.

Axonal growth cone dynamics involve the interplay of multiple protein components whose expression is known to be enriched within the growth cone, including receptors for chemo-attractants and -repellents that enable growth cones to recognize extracellular signals as well as cell adhesion factors, and extracellular matrix molecules that provide physical substrates through which cells can interact with their local environments[13]. Proteins belonging to the protein tyrosine phosphatase family mediate a wide variety of cellular activities and appear to play roles in cell-cell interactions, axon growth, and axon guidance[14]. The binding of one member of this family that is expressed in axonal growth cones, namely PTPσ, to chondroitin sulfate proteoglycans (CSPGs), extracellular matrix molecules, has been shown to inhibit axon elongation in injured neural tissues[15–18]. Conversely, the binding of heparan sulfate proteoglycans (HSPGs) to PTPσ attenuates the inhibitory effect of CSPGs, thus promoting growth cone extension and axon elongation in vitro[17].

In the postnatal brain, migrating neurons are produced in the ventricular-subventricular zone (V-SVZ), which lies along the ventricular wall. We have shown previously that neurons migrate from the V-SVZ toward injured sites[19] and that the location of migrated neurons is important for the recovery of gait function[20,21]. Given that CS inhibits neuronal migration[22,23], we hypothesized that the GCLS of migrating neurons may respond to the extracellular environment and guide neuronal migration in a manner that shares common features with axonal growth cone extension. If so, the receptors that regulate cytoskeletal remodeling, and thus neuronal migration, should be expressed in the GCLS of migrating neurons.

In this study, we investigated the cytoskeletal structure dynamics and molecular functions of the GCLS of migrating neurons and found that migrating neurons possess a growth cone that shares important functions with axonal growth cones. In brief, we found that the CSPG/HSPG receptor PTPσ is concentrated in migrating-neuron growth cones and regulates growth cone motility and neuronal migration. Functionally, we demonstrated that HSPGs promote neuronal migration induced by growth cone extension in an in vitro system mimicking CSPG-rich injured brain tissue as well as in injured mouse brain tissue with increased CSPGs. Using an artificial HSPG-containing scaffold, we succeeded in promoting growth cone-mediated neuronal migration, regeneration of mature neurons, and functional recovery in a mouse brain injury model. Based on these findings, we propose a neuronal migration mechanism wherein growth cones regulate migration through interaction with extracellular environments. Harnessing this mechanism could represent a potential strategy for restoring brain function after injury by modulating extracellular conditions.

## Results
### Migrating neurons possess a growth cone that resembles an axonal growth cone morphologically and molecularly

To compare the dynamics of GCLSs on the leading process (LPs) of migrating neurons with those of the axonal growth cones of differentiating neurons, we introduced a gene insert, Venus-CAAX[24], into cells of both types to enable visualization of cell membranes and then performed time-lapse super-resolution imaging (Fig. 1a, b; Supplementary Fig. 1a–f; Supplementary Movie 1). While the axons of cultured differentiating neurons derived from embryonic cerebral cortex elongated (Fig. 1a), cultured migrating neurons derived from the neonatal V-SVZ showed alternating LP elongation and pause phases with somal pausing and translocation, respectively (Fig. 1b, c), a pattern that resembles that exhibited by chain-forming migrating neurons in the rostral migratory stream (RMS)[25]. During the LP-elongation phase, GCLSs that resembled axonal growth cones morphologically and that had filopodia and lamellipodia were formed (Fig. 1a, b; Supplementary Fig. 1f). During the LP-pause phase, the GCLSs collapsed transiently (Figs. 1b, 21.5 min; Supplementary Fig. 1f). The phasic morphological changes exhibited by

migrating-neuron GCLSs resembled the axonal elongation processes of differentiating neurons.

Similar to axonal growth cones[5,26,27] the GCLSs on the LPs of migrating neurons contain filopodia and lamellipodia, both of which were confirmed to be enriched with F-actin and tyrosinated tubulin, the latter being a marker for dynamic microtubules (Fig. 1d, e; Supplementary Fig. 1g). Time-lapse imaging of EGFP-actin[28] and EB3-EGFP[29] (microtubule plus-end marker) revealed similar dynamics of F-actin and microtubules in both the GCLSs of elongating LPs and the growth cones of elongating axons[30] (Supplementary Fig. 1h–q; Supplementary Movie 1). These results indicate that the GCLSs on the LPs of migrating neurons are analogous to axonal growth cones in terms of both morphology and dynamics (Fig. 1a–e; Supplementary Fig. 1), and thus, for simplicity, we refer to them as LP growth cones from here forward.

We conducted immunocytochemistry analyses to examine whether molecules that we showed to be concentrated in axonal growth cones[31–35] are also concentrated in the LP growth cones (Fig. 1f, g). For this purpose, we used a machine learning-based algorithm to segment axons into growth cones and remaining areas in Dcx-EGFP-positive neurons (Fig. 1f). Applying this algorithm to Dcx-EGFP+ migrating neurons enabled us to demarcate the growth cones of LPs (Fig. 1g), consistent with the supposition that LP growth cones are analogous to axonal ones. We calculated the concentration of axonal growth cone molecules as the ratio of each immunopositive signal within identified LP growth cones relative to that in adjacent leading shafts. The CSPG/HSPG-responsive receptor PTPσ, the scaffold protein Liprin-α, and the actin depolymerizing factor Destrin were observed to be significantly concentrated in both LP and axonal growth cones (Fig. 1f, g; Supplementary Fig. 2a-e and 9a-d). PTPσ colocalized with F-actin in the peripheral domain of LP growth cones, including in filopodia and lamellipodia (Fig. 1h; Supplementary Fig. 2f). PTPσ was also found to be concentrated in the LP growth cones of migrating interneurons derived from embryonic ganglionic eminence (Supplementary Fig. 2h). These results indicate that concentrated expression of PTPσ is common to both axonal and LP growth cones. Liprin-α, a direct binding partner of PTPσ involved in PTPσ oligomerization[36], was found to be concentrated in LP growth cones and colocalized with F-actin (Fig. 1h). Liprin-α-knockdown (KD) reduced PTPσ signal levels in LP growth cones (Fig. 1i), suggesting that PTPσ concentration in LP growth cones is dependent on Liprin-α in migrating neurons.

To examine CSPG-PTPσ signaling effects on LP growth cone extension and subsequent neuronal migration, we analyzed how migrating neurons respond to the CSPG boundary in culture (Fig. 1j–p; Supplementary Fig. 2i). Similar to axonal growth cones[15,37,38], LP growth cones in contact with CSPG-containing Matrigel caused filopodia retraction and subsequent growth cone collapse [93.8 ± 6.3% ($n = 28$ events from 12 cells, four independent experiments)], followed by somal deceleration (Fig. 1k, l, o, p). This event was blocked by addition of chondroitinase ABC (ChABC), a CS-degrading enzyme, into the culture medium (Fig. 1m, o, p) or PTPσ-KD in migrating neurons (Fig. 1n–p), suggesting that CS-PTPσ signaling inhibits neuronal migration by restricting LP growth cone extension. Taken together, these results suggest that the LP growth cone of migrating neurons share morphological and molecular features with axonal growth cones.

### HSPGs promote LP growth cone extension and neuronal migration by relieving CSPG-PTPσ-mediated inhibition

PTPσ is a CSPG receptor that inhibits axon elongation in CS-rich injured neural tissues, and its inhibition promotes axonal regeneration[15,38]. In cultured axons, HS competition with CS enables PTPσ oligomerization and thus attenuates CS-mediated inhibition of growth cone extension[17], thereby promoting axonal elongation. To investigate whether HSPGs can attenuate CSPG-mediated inhibition of

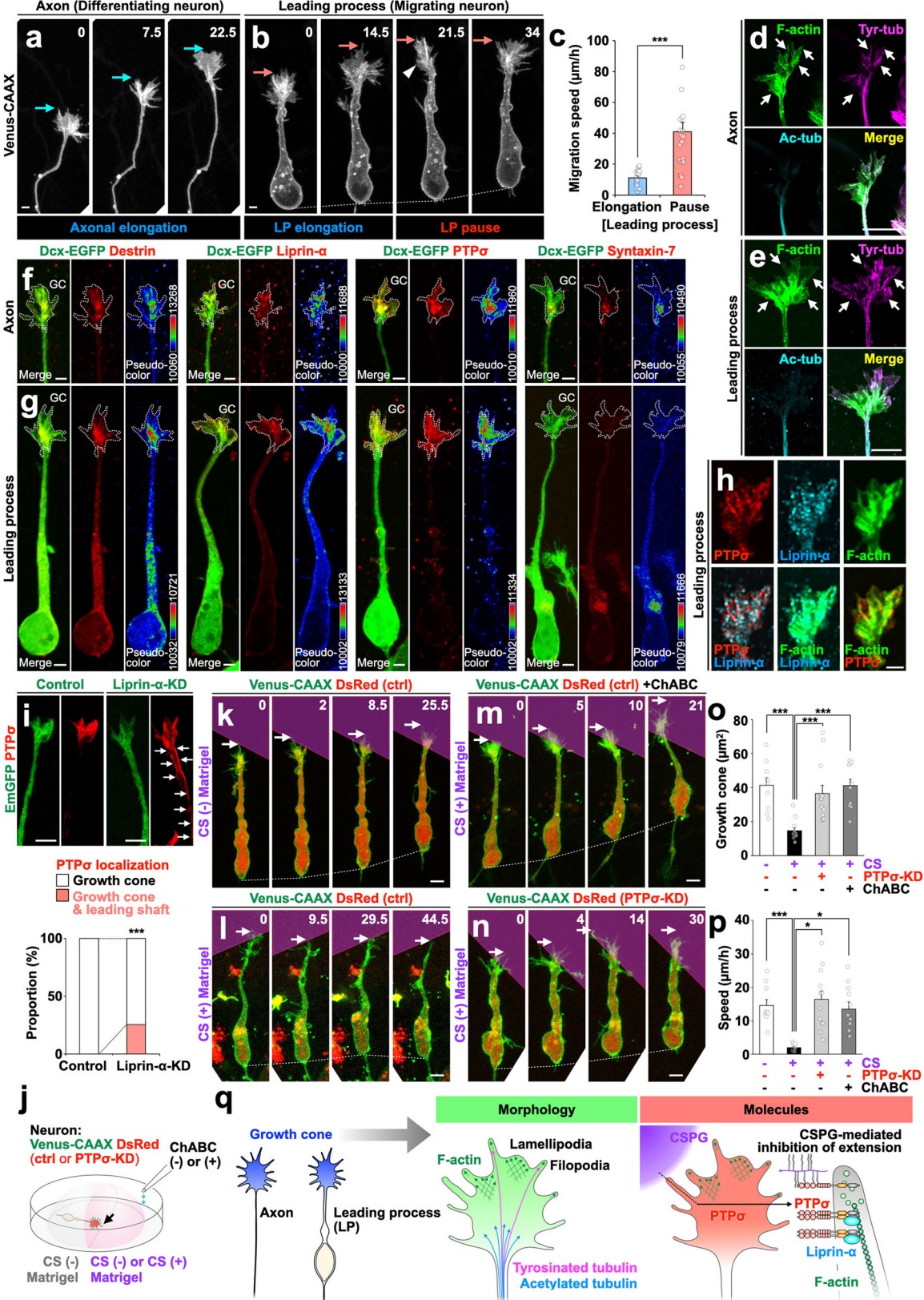

neuronal migration, we compared neuronal migration through CSPG-containing Matrigel in dishes coated versus not coated with the recombinant HSPGs glypican-1–6 (Gpc1–6) and syndecan-1–4 (Sdc1–4) and confirmed that HSPGs reduced CSPG-mediated inhibition of migration (Supplementary Fig. 3a–f). Addition of the HS-degrading enzyme heparinase-III reinstated CSPG-mediated inhibition of

migration (Supplementary Fig. 3b, c), suggesting that HS indeed enables neuronal migration in the presence of CSPGs in vitro, a condition mimicking that found in injured brain tissue.

We performed time-lapse super-resolution imaging of neurons migrating on dishes coated with stripes[39] of an HSPG, namely Sdc2, or a control protein [i.e., bovine serum albumin (BSA)](Fig. 2a–h;

**Fig. 1 | Leading process in migrating neurons shares morphological and molecular features with axonal growth cone. a, b** Time-lapse images of Venus-CAAX-labeled axon and leading process (LP) of differentiating (**a**) and migrating (**b**) neuron, respectively. Blue and pink arrows indicate tip of axonal growth cone and LP growth cone-like structure, respectively. Arrowhead indicates transient collapsed state of LP growth cone-like structure. **c** Migration speed in LP elongation and pause phases. **d, e** Representative images of an axonal growth cone (**d**) and LP growth cone-like structures (**e**) stained with phalloidin (green) and immunolabeled with antibodies targeting tyrosinated-tubulin (magenta) and acetylated-tubulin (cyan). Arrows indicate F-actin and tyrosinated tubulin-positive signals. **f, g** Representative super-resolution images of axonal (**f**) and LP (**g**) growth cones with labelling of GFP (green) and of the axonal growth cone molecules Destrin, Liprin-α, PTPσ, and Syntaxin-7 (red and pseudocolors [numbers indicate 16-bit depth]). Dotted lines indicate the segmented growth cone area (GC).

**h** Colocalization of PTPσ (red), Liprin-α (cyan), and F-actin (green) in the LP growth cone. **i** Representative super-resolution images of LP growth cones of control and Liprin-α-KD cells stained for GFP (green) and PTPσ (red). Arrows indicate PTPσ+ signals in the leading shaft. Quantification of PTPσ localization in the growth cone and leading shaft of control and Liprin-α-KD neurons is shown. **j** Experimental scheme for analysis of CSPG-PTPσ signaling. **k–p** Time-lapse images (**k–n**), growth cone area (**o**), and migration speed (**p**) of Venus-CAAX (green)- and DsRed (red; **k–m**, control; **n**, PTPσ-KD)-expressing migrating neurons cultured in Matrigel (magenta zone) with (**l–n**) or without **k** CS. Arrows indicate tips of growth cones (**k–n**). **q** Common morphological and molecular features of axonal and LP growth cones. Numbers indicate time in min (**a, b, k–n**) from the first imaging frame. Scale bars: **a, b, f–h**, 2 µm; **d, e, i, k–n**, 5 µm. *$p < 0.05$, ***$p < 0.005$. Error bars indicate mean ± SEM.

Supplementary Movie 2). Similar to axonal growth cones[38], LP growth cones collapsed and failed to extend through CSPG-containing Matrigel on control stripes (Fig. 2b, f, g) but each produced a singular filopodium on Sdc2 stripes (Fig. 2c, arrows) that showed PTPσ expression (Supplementary Fig. 2g). These tip filopodia on Sdc2 stripes were longer than filopodia on the control stripes (Fig. 2d). Following elongation of each tip filopodium, a lamellipodia-like section of plasma membrane extended in the same direction as the tip filopodium's elongation, forming a typical-appearing growth cone on Sdc2 stripes (Fig. 2c, 5–9 min; Fig. 2e) that moved anteriorly and was succeeded by somal translocation (Fig. 2c, 5–29 min; Fig. 2f, g). The maximum extension area of the LP growth cones correlated directly with cell migration speeds (Fig. 2h). Taken together, these results suggest that HSPGs induce tip filopodium formation in migrating neurons in the presence of CSPGs and promote neuronal migration by extending growth cones.

To examine the molecular mechanisms mediating growth cone extension and neuronal migration, we performed proteomic analyses of dissected cerebral cortex and RMS tissues from postnatal day 0 (P0), when differentiating and migrating neurons are enriched (Fig. 2i). A subset of proteins with similar relative proportions of expression in cortex and RMS was identified with the gene ontology terms growth cone and neuronal projection development (Supplementary Data 1). Thus identified candidate proteins included an actin-related factor (Cortactin), microtubule-related factors (Tctex-1, myosin heavy chain-10, and LIS1), and an RNA binding protein (HuD).

Cortactin phosphorylated at tyrosine 421 residue (pY421) is a PTPσ substrate in axonal growth cone[40]. Cortactin is a substrate of Src family kinases inclusive of Src[41] and Fyn[42]. It has been reported that Src2 targets Cortactin in axonal growth cones[43]. We showed that pY421-Cortactin is concentrated in LP growth cones (Fig. 2j) and tip filopodia (Supplementary Fig. 3g), and that its signal is reduced following the addition of PP2, a Src-family inhibitor (Supplementary Fig. 3h), or Fyn-KD (Fig. 2k), suggesting that Fyn influences pY421-Cortactin in LP growth cones. Moreover, Fyn- or Cortactin-KD caused defects in tip filopodium elongation, growth cone extension, and somal translocation of migrating neurons (Fig. 2l–n, q–s; Supplementary Fig. 3i and 9e), suggesting that Fyn and Cortactin are involved in these processes.

To examine the role of pY421-Cortactin in growth cone extension and neuronal migration, we introduced KD-resistant Cortactin (Cortactin*) and its Y421A mutant (Cortactin*Y421A) into Cortactin-KD neurons. The defects induced by Cortactin-KD were rescued by expressing Cortactin* but not Cortactin*Y421A (Fig. 2o–s; Supplementary Fig. 3i). These results suggest that pY421-Cortactin, whose activity is reciprocally regulated by Fyn and PTPσ, is critical for extension of tip filopodia and growth cones, and thus regulates neuronal migration in the presence of CS. Because the elongation of a tip filopodium in contact with Sdc2 leads to the extension of an associated growth cone under inhibitory conditions, hereafter we will refer to this filopodium as the leading filopodium (Supplementary Fig. 3k). The significance of leading filopodia was examined in the following experiments.

## Leading filopodia promote growth cone extension and subsequent neuronal migration in the presence of CS

Live imaging with fluorescently labeled actin and polymerizing microtubules (EGFP-actin and EB3-EGFP, respectively) confirmed that the elongating leading filopodia contain actin and microtubules (Supplementary Fig. 4a, b) similar to other filopodia of LP growth cones (Supplementary Fig. 1h–q). To elucidate the regulatory mechanisms of leading filopodium formation and its significance in growth cone extension, we used photoswitchable inhibitors of actin polymerization [Optolatrunculin (Opto-Lat)][44], F-actin depolymerization [Phenyl-*neo*-Optojasp (PnOJ), **1**, Supplementary Fig. 4c)][45], and microtubule polymerization [Photostatin-1 (PST-1), **2**, Supplementary Fig. 4c)][46]. These inhibitors are activated and inactivated by laser illumination at 405 nm and 514 nm, respectively, and thus can control cytoskeletal dynamics in a spatiotemporal manner. Leading filopodium tips on Sdc2 stripes were illuminated with a 405-nm laser to inhibit actin and microtubule dynamics, while the proximal LPs adjacent to the tips were illuminated with a 514-nm laser to suppress the activity of potentially diffusing activated inhibitors (Fig. 3a; Supplementary Fig. 4d–g).

The control laser illuminations, 405 nm without the inhibitors or 514 nm with the inhibitors, did not inhibit leading filopodium elongation (Fig. 3b, c, e, g; Supplementary Fig. 5a–d; Supplementary Movie 3), suggesting that laser illumination did not cause non-specific cell damage. Leading filopodium elongation into the 405 nm-illuminated region was hindered by inhibition of actin or microtubule polymerization with Opto-Lat and PST-1, respectively (Fig. 3c–g; Supplementary Movie 3). On the other hand, in the presence of the F-actin depolymerization inhibitor PnOJ (activated), the leading filopodium was maintained, suggesting that F-actin depolymerization is involved in leading filopodium retraction (Supplementary Fig. 5a–d). These results suggest that actin and microtubules regulate the dynamics of leading filopodia.

Finally, we analyzed the role of the leading filopodium on growth cone extension and neuronal migration (Fig. 3c–i). Neurons that could not elongate the leading filopodium failed to develop lamellipodia in the 514 nm-illuminated region, resulting in suppression of growth cone extension and neuronal migration (Fig. 3d, f, h, i; Supplementary Movie 3). Inhibition of the formation of non-leading filopodia did not inhibit growth cone extension (Supplementary Fig. 5e–h). These results suggest that the leading filopodium plays an important role in growth cone extension and neuronal migration in the presence of CS (Supplementary Fig. 3k).

## Migrating neurons in CS-rich injured cortex collapse their growth cones

The in vitro experiments showed that growth cone dynamics are important for neuronal migration. We next performed histological analysis in a mouse model of brain injury to examine whether in vivo neuronal migration employs the growth cone regulatory mechanisms that we observed in vitro. Cortical cryoinjury was induced in the brains

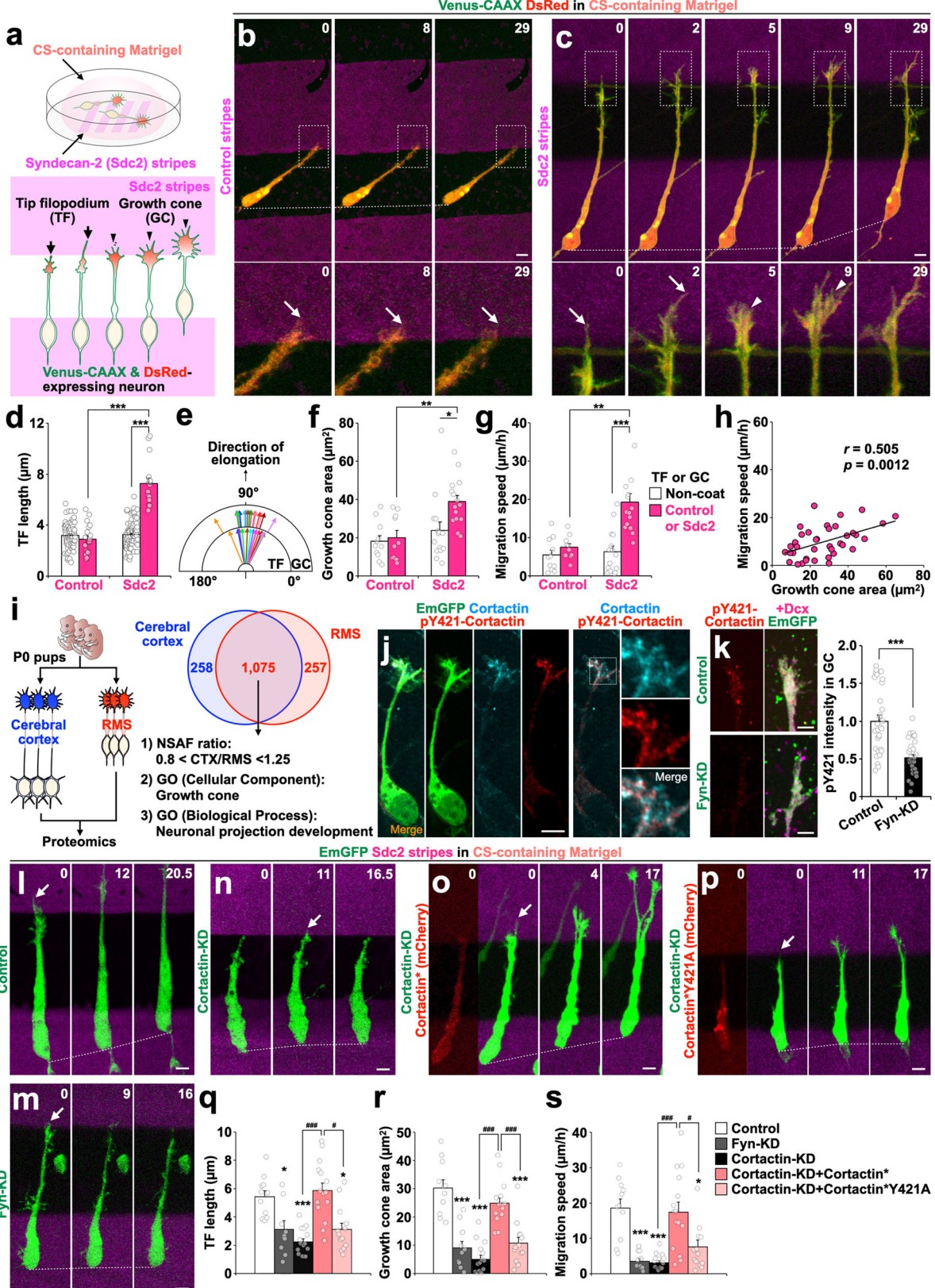

of P2 *Dcx-EGFP* mice, and CS signal intensity 7 d after injury was focused near the injury core (Fig. 4a, b; Supplementary Fig. 6a). V-SVZ neural stem cell-derived neurons migrated toward the CS-enriched injured site (Fig. 4b–d).

Neurons migrating in the injured cortex expressed PTPσ in their growth cones similar to cultured neurons (Fig. 4e). Percentages of

migrating neurons with membrane-extended growth cones, defined as having a width at least twice that of the leading shaft, were determined. Neurons in the CS-enriched region formed less-extended growth cones than neurons near the dorsal V-SVZ (Fig. 4f, g). Serial block-face scanning electron microscopy (SBF-SEM) analysis showed that enlarged growth cones in the injured cortex exhibited extended

**Fig. 2 | Heparan sulfate proteoglycans promote growth cone extension and neuronal migration by relieving chondroitin sulfate-PTPσ-mediated inhibition. a** Experimental design for Sdc2-coated stripe assay embedded in the CS-containing Matrigel. **b, c** Time-lapse images of Venus-CAAX (green)- and DsRed (red)-expressing migrating neurons on control or Sdc2 stripes (magenta) cultured in the CS-containing Matrigel. Boxed areas are enlarged at the bottom. Arrows and arrowheads indicate tip filopodium and growth cone, respectively. **d** Length of tip filopodium on control and Sdc2 stripes. **e** Degree of formed tip filopodium (TF) and growth cone (GC) on Sdc2 stripes. **f, g** Growth cone area (**f**) and migration speed (**g**). **h** Correlation between growth cone area and migration speed. **i** Scheme of proteomics analysis. **j** Localization of Cortactin (cyan) and its pY421 form (red) in an LP growth cone of an EmGFP+ (green) neuron. Boxed area is enlarged. **k** Localization (left) and intensity (right) of pY421-Cortactin (red) in the growth cones of control and Fyn-KD (EmGFP, green) Dcx+ (magenta) neurons. **l–s** Functional analyses of Fyn and Cortactin in neuronal migration. Time-lapse images of EmGFP (green)-expressing migrating neurons on Sdc2 stripes (magenta); the neurons were cultured in CS-containing Matrigel. KD-resistant WT- (**o**) or Y421A- (**p**) Cortactin (Cortactin*) expressing cells are labelled with mCherry (red). Tip filopodium length (**q**), growth cone area (**r**), and migration speed (**s**) are shown. Arrows indicate tip filopodium on Sdc2 stripes. Numbers indicate time in min (**b, c, l–p**) from the first imaging frame. Scale bars: **b, c, j, l–p**, 5 µm; k, 2 µm. *$p < 0.05$, **$p < 0.01$, ***$p < 0.005$ (in **q–s**, vs Control); #$p < 0.05$, ###$p < 0.005$. Error bars indicate mean ± SEM.

filopodia and lamellipodia (Fig. 4h; Supplementary Movie 4), while smaller growth cones were associated with membrane retraction (Fig. 4i; Supplementary Movie 4). Furthermore, in organotypic brain slice cultures from cryoinjured *Dcx-EGFP* mice, EGFP-positive neurons migrated through the injured site with LP elongation and repeated expansion and retraction of growth cones (Supplementary Movie 5). Some neurons had LPs that changed direction in the injured sites (Fig. 4j; Supplementary Movie 5). Hence, we observed neurons migrating in vivo with repeated growth cone extension and retraction, and we found that neurons in and around CS-rich injured sites showed inhibited growth cone extension and hindered migration.

## HSPG-containing biomaterial serves as a scaffold for migrating neurons in the injured brain

Given the involvement of proteoglycans, PTPσ, and Cortactin in regulating growth cone extension, we hypothesized that modulating the extracellular environment in injured cortex may affect neuronal migration by altering growth cone dynamics. To examine whether HSPG-containing substrate promotes neuronal migration into injured cortex, we employed gelatin-fiber nonwoven fabric (GF), a unique biomaterial that provides a structural scaffold for cells and has a high affinity for extracellular matrix molecules, including sulfated glycosaminoglycan[47,48] (Fig. 5a). GF consists of parallel long gelatin fibers with randomly-interspersed perpendicularly-oriented short fibers for support (see X-ray computed tomography in Fig. 5b–c; Supplementary Movie 6) and sufficient inter-fiber spaces (Fig. 5d; Supplementary Movie 6; void = 63.8% v/v) for neurons to migrate radially through the fibrous structure. GFs subjected to two thermal cross-linking conditions were compared by implanting them in the injured cortices of P5 neonatal mice, 3 d after cryoinjury. Four days after implantation, the GF cross-linked at a higher temperature (160 °C, 24 h) maintained larger inter-fiber spaces than the GF cross-linked at a lower temperature (140 °C, 12 h) (Supplementary Fig. 6b, c). Moreover, 25 d after implantation, the former had maintained more gelatin fibers than the latter (Supplementary Fig. 6d) and thus was selected as the more suitable scaffold material for migrating neurons.

Time-lapse imaging showed that cultured neurons migrated faster on GF that had been loaded (by way of immersion) with Sdc2, Sdc4, or Gpc4 (the HSPGs used in the above experiments) than on control GF without the HSPG (Fig. 5e–g; Supplementary Fig. 6e, f; Supplementary Movie 7). Next, we conducted in vivo experiments in injured cortex with HSPG-loaded GF that had been cross-linked at 160 °C for 24 h. GFs were implanted vertically such that the long gelatin fibers facilitated radial migration (Supplementary Fig. 6g, h). Immunohistochemical analysis of brain sections with implanted GFs revealed that the densities of Dcx-EGFP–positive neurons in the middle and upper layers were higher in the injured cortex implanted with HSPG-containing GF than those in the injured cortex implanted with control GF (Fig. 6a). More detailed analyses showed that the frequency and density of neurons along the gelatin fibers per se were higher in HSPG-containing GF than in non-HSPG control GF (Fig. 6b–d), and that the cell bodies of migrating neurons were in direct contact with Sdc2-bound gelatin

fibers (transmission electron microscopy images, Fig. 6e, f). These data indicate that HSPG-enhanced gelatin fibers can serve as direct scaffolding that facilitates neuronal migration.

We compared neuronal migration in injured brain implanted with Sdc2-loaded GF, Sdc2-enriched polypropylene-fiber nonwoven fabric (PF)[47] or gelatin sponge. Dried gelatin sponge has a honeycomb-like structure consisting of gelatin film walls with less interconnected pores, while the dried PF has a lattice-like structure of fibers similar to that of GF (scanning electron microscopy images, Fig. 6g). Dcx-EGFP+ cell density was greater in cortices implanted with Sdc2-enriched GF than in those implanted with Sdc2-enriched PF or gelatin sponge (Fig. 6h; Supplementary Fig. 6i), suggesting that GF structure and material are more conducive to neuronal migration.

Slice culture experiments were performed to analyze the kinetics and migration velocity of neurons in injured brain sections implanted with GF. Sdc2-loaded and non-loaded GFs were labeled with Dye-Light655 and implanted into injured cortex sites in *Dcx-EGFP* neonatal mice. It was possible to recognize the growth cone morphology from the resultant brain sections (Supplementary Fig. 7a). Greater numbers of migrating neurons were observed along Sdc2-loaded gelatin fibers than along non-loaded fibers (Fig. 6i; Supplementary Fig. 7b; Supplementary Movie 8). When an EGFP+ leading filopodium (arrow) made contact with the Sdc2-containing scaffold, LP growth cone extension occurred (arrowheads, after which the cell body moved toward the swelling (asterisk) (Fig. 6j; Supplementary Fig. 7c; Supplementary Movie 8). Quantitative analyses of neuronal activities upon scaffold contact revealed that, compared to neurons in non-loaded GF, neurons in Sdc2-containing GF extended their growth cones more frequently (Fig. 6k), had a higher probability of migration after growth cone extension (Fig. 6l), and migrated faster (Fig. 6m). Additional quantitative analysis of slice culture images of injured cortex implanted with Gpc4- or Sdc4-loaded GFs revealed significantly more frequent growth cone extension after contact with Gpc4- or Sdc4-enriched GFs than after contact with control GF (Supplementary Fig. 7d, e; Supplementary Movie 9). These results indicated that HSPG-augmented GF is an excellent scaffold for promoting LP growth cone extension and neuronal migration.

## HSPG-containing GF scaffold facilitates neuronal regeneration with functional recovery after cortical injury

To examine the maturation of regenerated neurons and functional recovery, we labeled V-SVZ cells in the lateral walls of the lateral ventricles of mouse pups by electroplating EmGFP-encoding plasmid at P0. We then proceeded with inducing cortical injury, GF implantation, gait function testing, and immunohistochemical analyses of the resulting P30 brain sections (experimental overview in Fig. 7a). Significantly larger numbers of EmGFP+NeuN+ mature neurons were observed in brains implanted with HSPG-loaded GF than in brains with control GF (Fig. 7b–f; Supplementary Fig. 7f). Foot-fault test analysis indicated that mice with the HSPG-containing scaffolds had significantly better gait function than mice implanted with the control GF or mice subjected to cryoinjury without scaffold implantation (Fig. 7g).

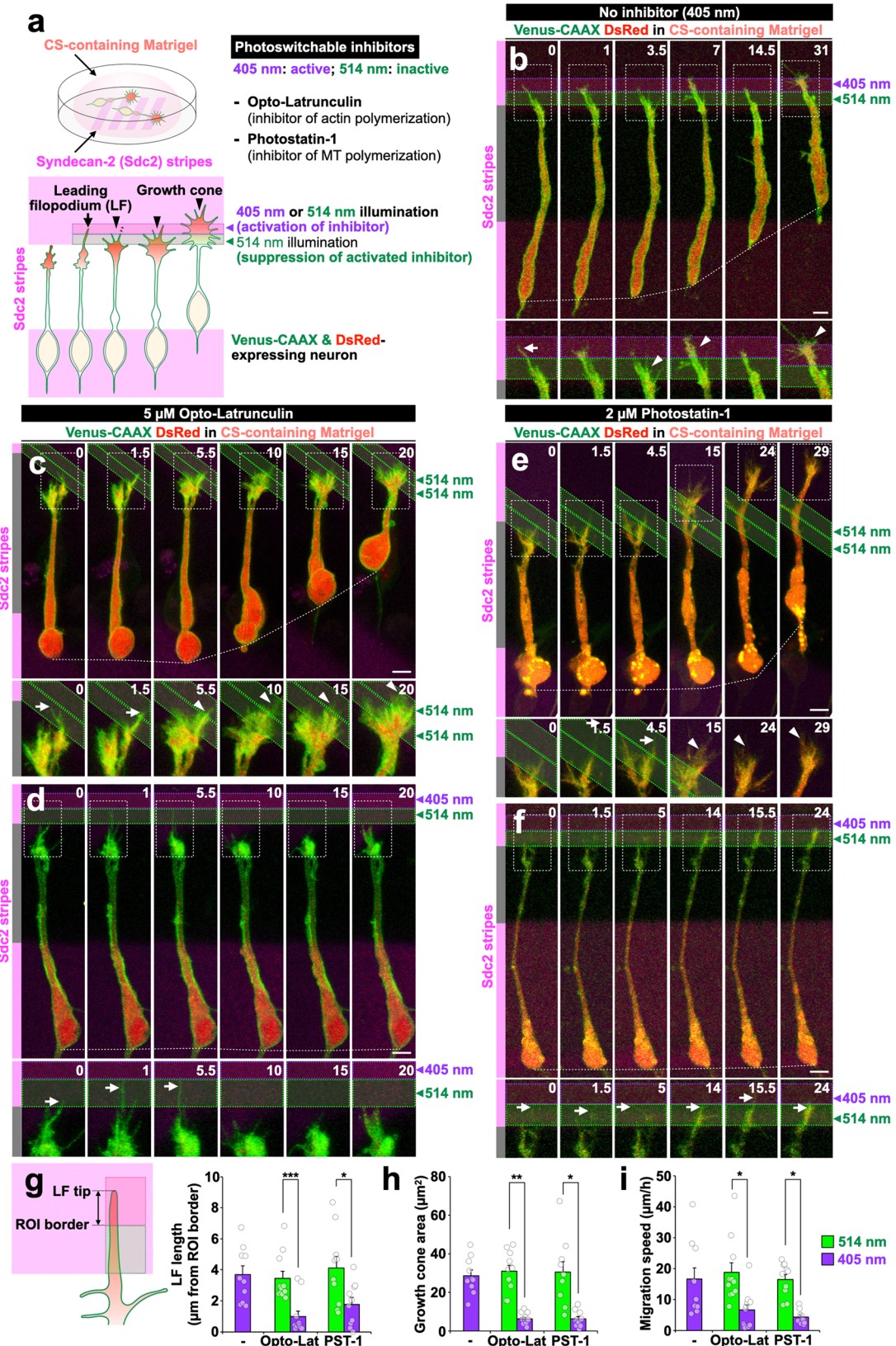

**Fig. 3 | Leading filopodium promotes growth cone extension and subsequent neuronal migration in the presence of CS. a** Experimental design using Opto-Latrunculin and Photostatin-1. **b–f** Time-lapse images of Venus-CAAX (green)- and DsRed (red)-expressing migrating neurons on Sdc2 stripes (magenta) cultured without inhibitors (**b**), with 5 µM Opto-Latrunculin (**c, d**), or with 2 µM Photostatin-1 (**e, f**). Regions of illumination with 514 (**b–f**) and 405 (**b, d, f**) nm laser are shown in green and purple box, respectively. **g–i** Leading filopodium length (**g**), growth cone area (**h**), and speed (**i**) of migrating neurons treated with Opto-Latrunculin (Opto-Lat) or Photostatin-1 (PST-1). Arrows and arrowheads indicate leading filopodium and growth cone, respectively. Numbers indicate time in min from the first imaging frame (**b–f**). Scale bars: **b–f**, 5 µm. *$p < 0.05$, **$p < 0.01$, ***$p < 0.005$. Error bars indicate mean ± SEM.

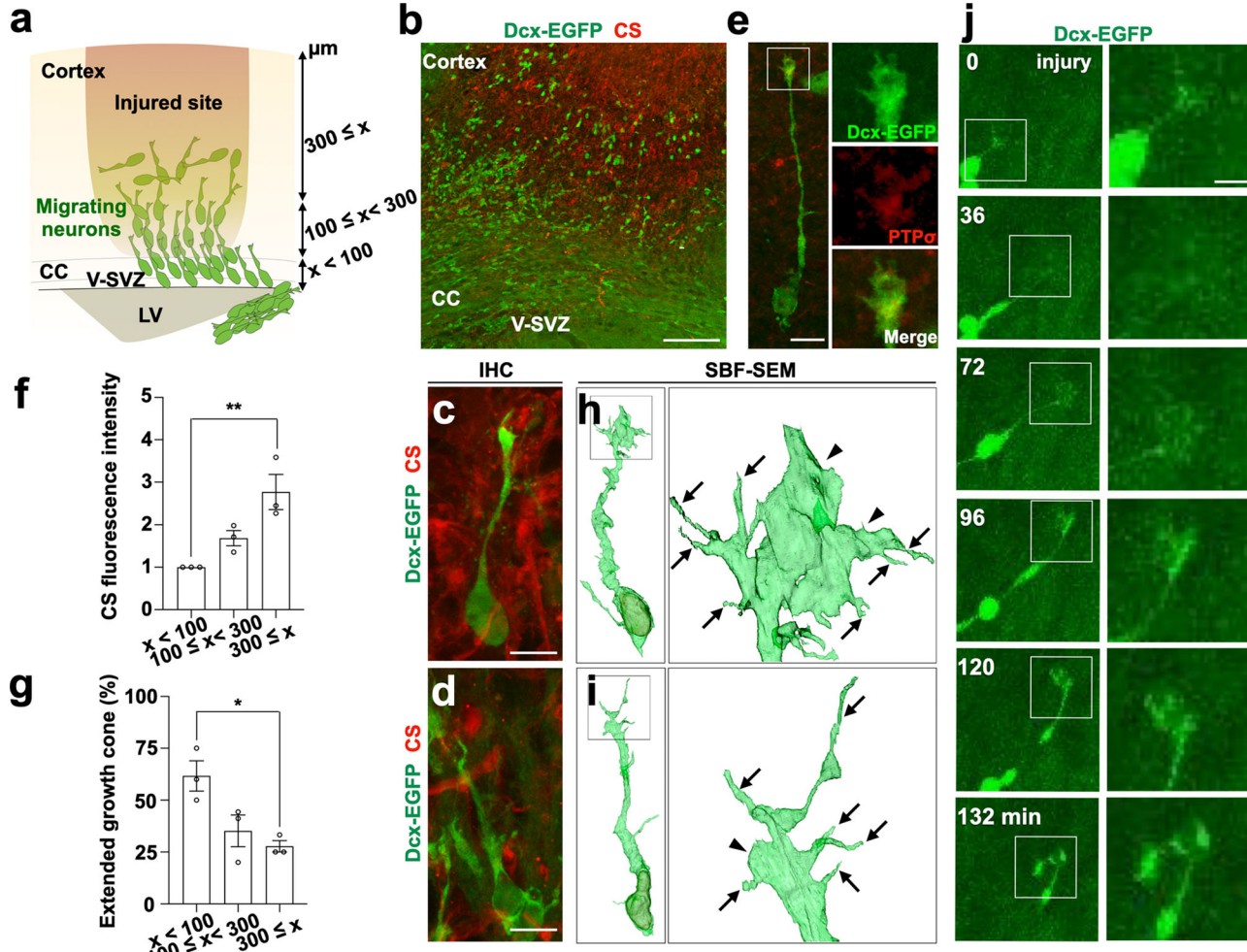

**Fig. 4 | Migrating neurons in CS-rich injured cortex collapse their growth cones. a** Schematic image shows coronal brain section with neurons (green) migrating from the V-SVZ toward the injured cortical layers (brown) through the corpus callosum (CC). The scale indicates the distance from dorsal V-SVZ used in (**d**) and (**e**). **b–d** A Z-stack projection image of Dcx-EGFP+ cells (green) and CS (red) in a P9 injured brain section including V-SVZ, corpus callosum (CC), and cortex. The representative extended growth cone morphology (**c**) and collapsed morphology (**d**) of Dcx-EGFP+ cells in CS-induced injured cortex. **e** A representative image of Dcx-EGFP+ cells (green) expressing PTPσ (red) in the P14 injured cortex. The boxed area is enlarged. **f** Relative intensity of CS signal in the P9 injured cortex (0 to <100 μm, 100 to <300 μm, and 300 ≤ μm from the dorsal V-SVZ) was analyzed. The intensity in the area within 100 μm from the dorsal V-SVZ was set as 1. **g** The percentage of extended growth cones of observed Dcx-EGFP+ cells in the P9 injured cortex (0 to

<100 μm, 100 to <300 μm, and 300 ≤ μm from the dorsal V-SVZ). **h, i** Representative three-dimensional constructions of migrating neurons (green) in the P9 injured cortex analyzed by SBF-SEM. Yellow represents nuclei. Extended (**h**) and collapsed (**i**) growth cones are shown. Boxed areas are enlarged. Arrows and arrowheads indicate filopodia and lamellipodia, respectively. Interactive 3D models of neurons are shown at https://sketchfab.com/3d-models/migratory-neuron-with-extended-growth-cone-b6c4b616f56343929cab8e3edca1c884 (**h**) and https://sketchfab.com/3d-models/migratory-neuron-with-collapsed-growth-cone-70648b036df64a01a30339717b22537f (**i**). **j** Time-lapse images of a slice cultured P8 *Dcx-EGFP* injured cortex section. A Dcx-EGFP+ cell migrates towards the injured site and the growth cone changes direction during the migration. Boxed areas are enlarged and shown on the right panels. Scale bars: **b**, 100 μm; **c**, **d**, **e**, 10 μm; **j**, 5 μm. *p < 0.05, **p < 0.01. Error bars indicate mean ± SEM.

Finally, adenovirus encoding Cre-recombinase was injected into the ventricles of P0 neuron-specific enolase-diphtheria toxin fragment A (*NSE-DTA*) mice[49,50], in which postnatally generated neurons derived from the V-SVZ are ablated (Fig. 7h). Implantation of Sdc2-loaded GF did not restore gait function in cortex-cryoinjured *NSE-DTA* mice (Fig. 7i). These results suggest that application of HSPG-containing scaffolds to an injured brain can promote neuronal migration, thereby enabling neuronal maturation and facilitating recovery of brain function (Supplementary Fig. 8).

## Discussion

Postnatal neuronal migration[51,52] is crucial for acquisition of brain functions and for functional recovery after brain injury[53]. Though many intrinsic and extrinsic mechanisms regulating neuronal migration have been described[53], the role of tip structures of migrating neurons has

not been clarified. The present in vitro and in vivo findings demonstrate that a growth cone exists in migrating neurons and its activities are indispensable for neuronal migration, especially in inhibitory environments. Thus, the findings of this study allow us to propose a promising strategy for facilitating neuronal migration in the injured brain.

We first demonstrated that the expression and distribution of cytoskeletal proteins are highly similar between the axonal and LP growth cones. During the elongation of both LPs and axons, F-actin and microtubule polymerized at the plus-end and showed similar growth cone dynamics. We showed that, like their axonal counterparts, LP growth cones express PTPσ and collapse in response to CSPGs[15,17,38], which are abundant in the injured brain. Importantly, LP growth-cone extension correlated with migration speed and was consistent with functions previously described for axonal growth cones[17], with

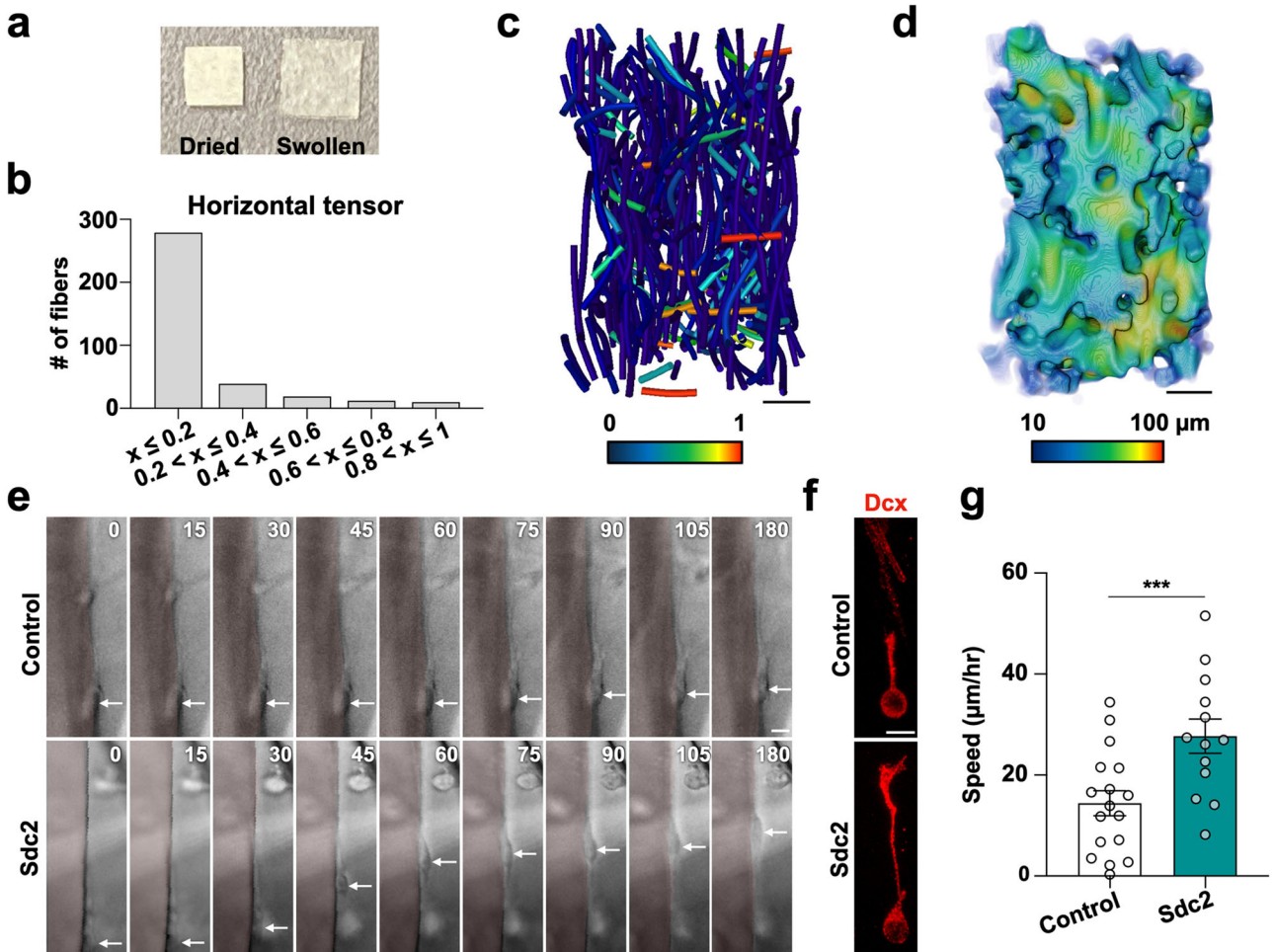

**Fig. 5 | Gelatin-fiber nonwoven fabric serves as a scaffold for migrating neurons. a** Dried and swollen gelatin fabrics. **b, c** The alignment distribution of gelatin fibers (**b**) and fiber centerlines (**c**). The color-code indicates the horizontal fiber orientation tensor (the more horizontal the fiber, the higher the value). **d** 3D image of voids among gelatin fibers in the fabric. The color-code indicates the void thickness. **e** Time-lapse images of migrating neurons (arrows) aligned with the control or Sdc2-containing gelatin fiber (pale pink) at different time points (minutes). **f** Representative Z-stack confocal images of migrating neurons (Dcx; red) on the gelatin fibers with or without Sdc2. **g** The speed of neurons migrating along the control or Sdc2-containing gelatin fibers embedded in Matrigel with CSPG. Scale bars: **c, d**, 100 µm; **e, f**, 5 µm. ***$p < 0.005$. Error bars indicate the mean ± SEM.

attenuation of CSPG inhibition enabling LP growth-cone extension and neuronal migration. It is likely that LP growth cones have downstream mechanisms in common with axonal growth cones, particularly those involving Liprin-α[36] and Cortactin[40]. We have focused on Cortactin, Destrin, Liprin-α, and PTPσ as concentrated molecules in the LP growth cone. Notwithstanding, there could be other yet-to-be-examined growth-cone concentrated molecules or even non-concentrated proteins that are important mediators of LP growth cone functions. Recent studies have implicated ATP, cAMP, and $Ca^{2+}$ in the LP-soma dynamics of migrating neurons[54–56]. Thus, growth cone dynamics controlled by PTPσ-mediated cytoskeletal reorganization may regulate saltatory neuronal migration by affecting these intracellular signaling molecules. Furthermore, the common expression of PTPσ within the growth cone of migrating neurons derived from multiple origins (Supplementary Fig. 2) is suggestive of PTPσ involvement in mechanisms regulating neuronal migration in general. The present findings indicate that neuronal migration is mediated by functional growth cones whose activities are regulated by extracellular and intracellular molecules through PTPσ.

Typically, axonal growth cones are motile owing to dynamic filopodia and lamellipodia with a high rate of actin turnover in their leading edges and collapse in the presence of CS, an activity blocker. In this study, we found that the activities of the growth cone of migrating neurons resemble those of axonal growth cones. We further observed a phenomenon not previously described for axonal growth cones. That is, upon contacting HSPG in the presence of CS, each LP growth cone retains a singular filopodium at the tip of the leading edge. This filopodium, which we describe as the leading filopodium in this study, persists as a minimal sensory-motile structure. Contact with HSPG in a CS-containing environment was followed by leading filopodium elongation, lamellipodia extension, LP growth cone extension, and, ultimately, somal translocation. Our photoinactivation experiments suggested that proper spatiotemporal regulation of F-actin and microtubule assembly dynamics is essential for leading filopodium elongation and further suggested that lamellipodia extension and cell body movement are strongly dependent on this process. Our observation of PTPσ expression in the leading filopodium together with the aligned directionality of the leading filopodium and LP growth cone are consistent with the possibility that the leading filopodium may sense appropriate environments for neuronal migration and thus be involved in growth cone guidance.

Inhibition of the formation of non-leading filopodia did not restrict LP growth cone extension, suggesting that the leading filopodium is a structure that plays a pivotal role in growth cone extension in CSPG-rich inhibitory environments. Impaired mitochondrial respiration and glycolysis in the presence of CSPGs, which reduces ATP

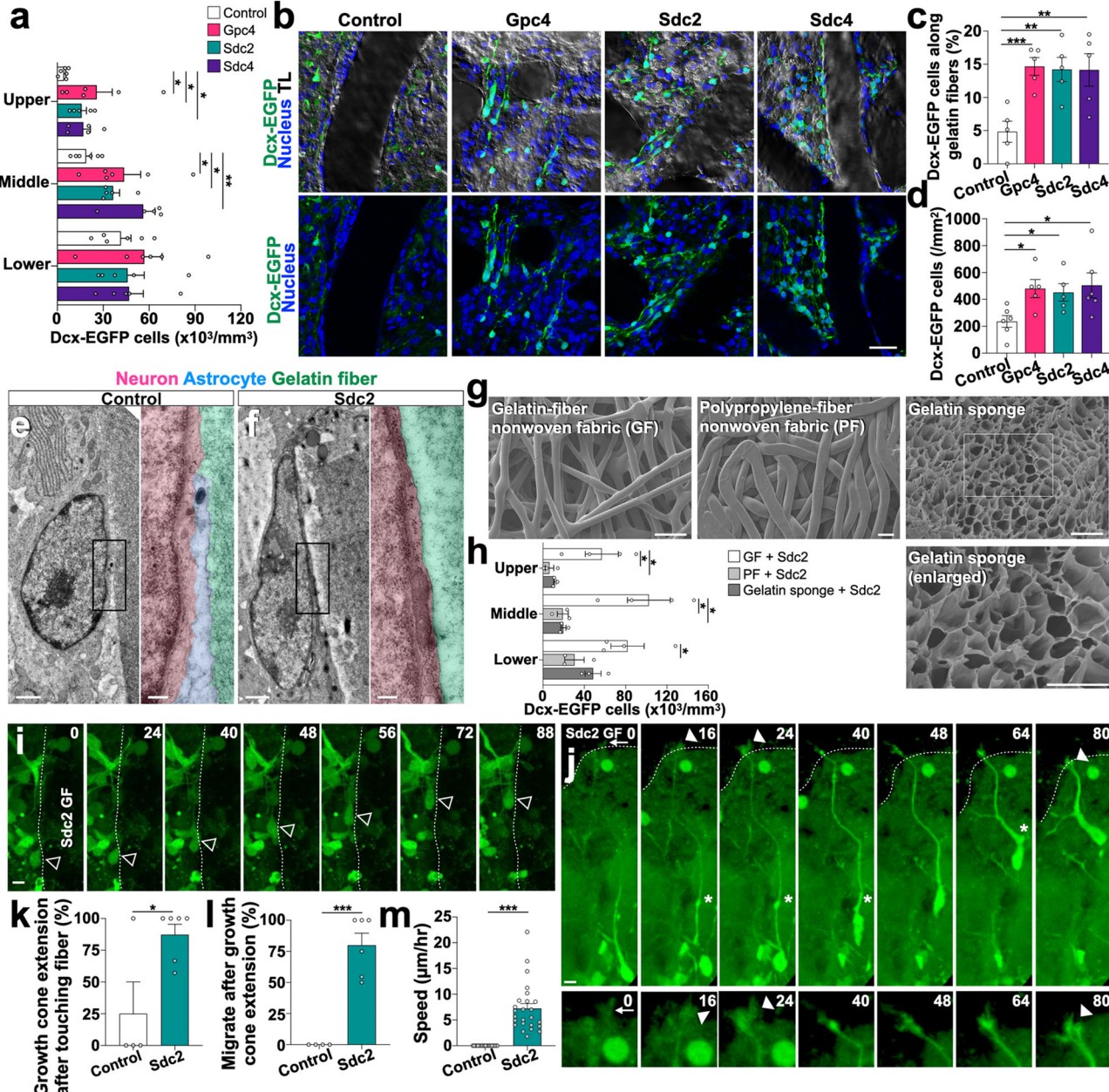

**Fig. 6 | Gelatin fabrics containing HSPG promote growth cone extension and neuronal migrating in the injured brain. a** Densities of Dcx-EGFP+ neurons in the P9-injured cortical layers (lower, middle, and upper) containing gelatin fabrics with or without HSPGs. **b** Distribution of Dcx-EGFP+ neurons in the vicinity of and along the gelatin fibers with or without HSPGs in the P9-injured cortex. Transmitted light images (TL) are overlaid with Dcx-EGFP+ cells (green) and Hoechst 33342 (blue) Z-stack projection images. **c**, **d** The percentage (**c**) and density (**d**) of Dcx-EGFP+ cells along the gelatin fibers in P9 middle and upper cortex. **e**, **f** Transmission electron microscopy of the P9 cryoinjured cortex with implanted gelatin fibers. Boxed areas are enlarged. Migrating neurons (pink), gelatin fibers (green), and an astrocyte (blue). **g** Scanning electron microscopy of dried biomaterials. The cross-section image of the gelatin sponge is enlarged. **h** Density of Dcx-EGFP+ neurons in P9-injured cortical layers (lower, middle, and upper) implanted with Sdc2-including gelatin fabric (GF), gelatin sponge, and polypropylene fabric (PF). **i**, **j** Time-lapse images of *Dcx-EGFP* brain slice cultures with Sdc2-containing gelatin fibers. A Dcx-EGFP+ cell (open arrowheads) migrates along the fiber (**i**). Another cell migrates a long distance toward the fiber (**j**). An arrow, arrowheads, and asterisks indicate a filopodium, extended growth cones, and swellings, respectively. Dotted lines indicate the borders between the gelatin fiber and the tissue. **k**–**m** The graphs show the percentage of growth cone extension after touching gelatin fibers (**k**), the percentage of neuronal migration within 2 hours after growth cone extension (**l**), and the speed of migrating neurons attached to gelatin fibers (**m**). Scale bars: **b**, 200 μm; **g** (GF and PF), 100 μm; **g** (gelatin sponge), 10 μm; **i**, **j**, 5 μm; **e**, **f**, 1 μm; **e** (enlarged), **f** (enlarged), 0.2 μm. *$p < 0.05$, **$p < 0.01$, ***$p < 0.005$. Error bars indicate the mean ± SEM.

production and limits cytoskeletal remodeling[57–59], may prevent simultaneous elongation of multiple filopodia, leaving a single leading filopodium that can achieve sufficient elongation for environmental sensing.

HSPG and CSPG exert opposing effects on axon growth by competing on PTPσ[17]. We demonstrated here using in vitro and in vivo systems that HSPGs directly attenuate the inhibitory effect of CSPGs on LP growth cone activities and promote neuronal migration. Indeed, HSPG-containing biomaterials promoted extension of the LP growth cone and migration of neurons in the injured regions. Furthermore, the migrated neurons matured in the injured brain, and their maturation was associated with the recovery of neurological function.

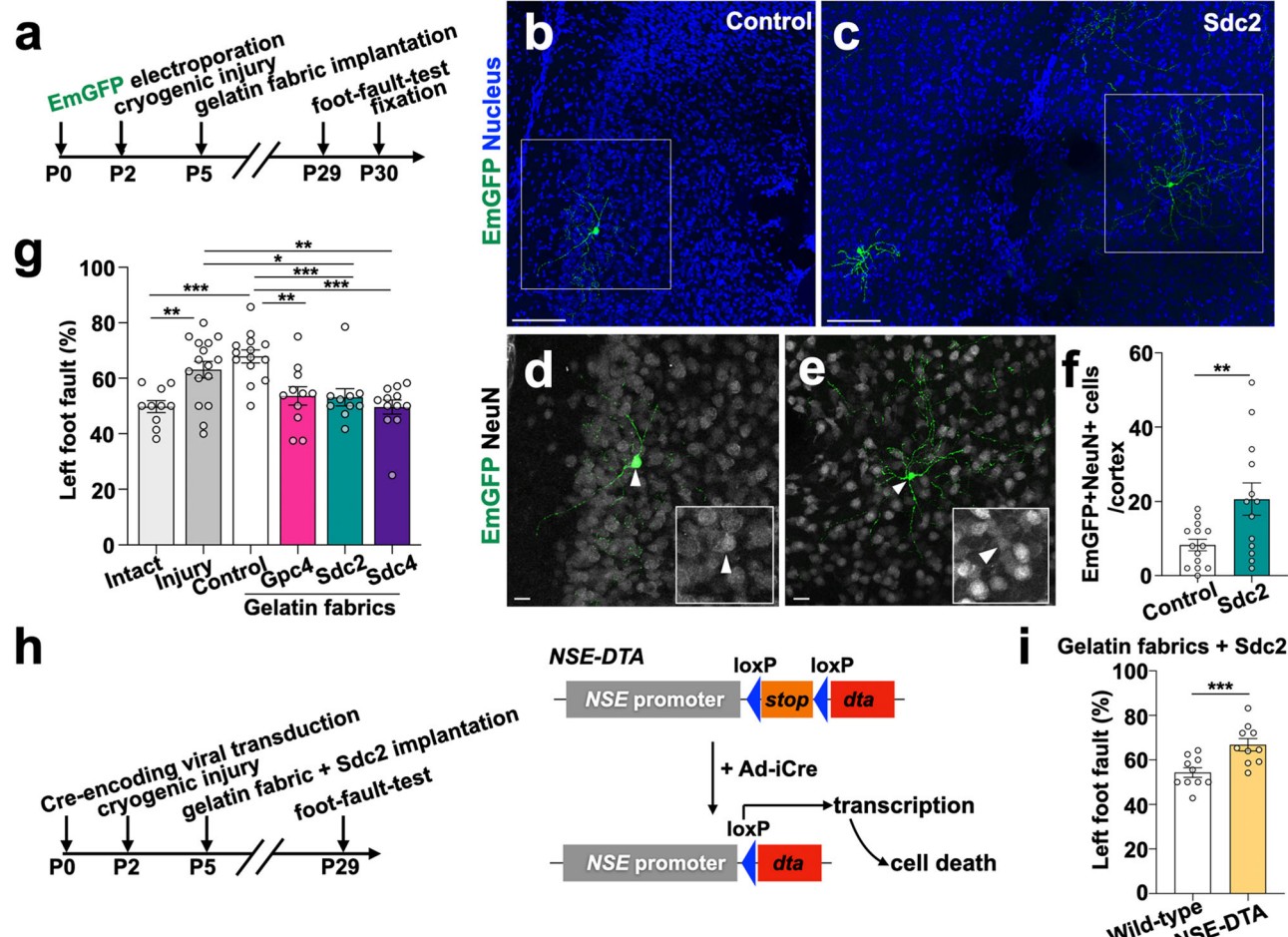

**Fig. 7 | HSPG-containing gelatin scaffold facilitates neuronal regeneration with functional recovery after cortical injury. a** Experimental design for assessment of neuronal maturation and gait behavior. **b–e** Z-stack projection images of coronal brain sections show that EmGFP-labeled (green) V-SVZ-derived cells in the vicinity of the injury express mature neuronal marker NeuN (white). Magnified views of the EmGFP+NeuN+ cells in the boxed areas are presented in (**d** and **e**) (white arrowheads). **f** Quantification of EmGFP+NeuN+ cells in injured cortex. The graph shows the total number of EmGFP-labelled NeuN+ cells in P30 control or Sdc2-containing gelatin fabric-implanted injured cortices. A dot represents the examined brain. **g** Foot-fault test in mice with or without cortical injury and gelatin fabric implant. The graph represents the frequency of left hindlimb fault-steps. **h** Experimental design for the experiment using *NSE-DTA* mice. **i** Foot-fault test on adenovirus expressing Cre recombinase-injected *NSE-DTA* mice and wild-type mice, implanted with Sdc2-including gelatin fabric. The graph represents the frequency of left hindlimb fault-steps. Scale bars: **b**, **c**, 100 μm; **d**, **e**, 10 μm. *$p < 0.05$, **$p < 0.01$, ***$p < 0.005$. Error bars indicate mean ± SEM.

Previous studies have shown that local administration of the bacterial enzyme ChABC[60] or subcutaneous injection of an intracellular signal peptide, a cell-permeable PTPσ inhibitor[23], reduces the repulsive effects of CS and restores brain function after stroke. While those studies aimed to reduce the inhibitory effects of CSPG in the injured brain tissue, our study uniquely demonstrated an association between the functional recovery and the extension of the LP growth cone and subsequent neuronal migration. Thus, the present findings provide a fundamental insight into the process of neuronal migration and the facilitation of functional recovery.

Regarding biomaterials, compared to gelatin-sponges with randomly oriented[20] and less interconnected pores (Fig. 6), gelatin-fabrics can better promote radial neuronal migration toward upper cortical layers because they consist of aligned fibers. Notably, embryonic radial glial cells express HSPGs[61] and migrating neurons in both embryonic and postnatal injured brains use radial glial cells as scaffolds for migration[20,62]. To mimic the broad distribution of radial glial fibers in the developing cortex, it would be preferable to implant a biomaterial that covers a large area rather than to inject biomaterial such that it affects a small area around the injection site, though the latter approach has the advantage of minimizing the invasiveness of the

procedure[63,64]. Thus, HSPG-augmented gelatin-fabrics mimic endogenous scaffolds both structurally and functionally. Biomaterials containing cell adhesion molecules, such as N-cadherin[20,64] and laminin[63,65], may promote neuronal migration as artificial scaffolds only when the cell bodies of migrating neurons reach the material. In this study, we observed very long LPs of neurons that responded to HSPGs in scaffold material and thus promoted neuronal migration over long distances in response to HSPGs being detected by their LP growth cones far from the cell body. Therefore, HSPG-containing gelatin-fabrics have exceptional potential to promote neuronal migration even when implanted at a site distant from the V-SVZ, thus avoiding implantation-induced brain damage.

This study demonstrated the existence and functionality of the growth cone of migrating neurons and provided evidence for molecular mechanisms underlying that migration, thus contributing to our understanding of how neuronal migration is regulated in response to the surrounding extracellular mileau[66], the composition of which evolves during development and is altered in adult brains by disease or injury. This study further demonstrated that manipulating the regulatory mechanisms of LP growth cones can promote neuronal migration in injured brains and restore brain function. Elucidation of

the molecular mechanisms that mediate growth cone interactions with the local extracellular environment and thereby promote growth cone extension will enable the development of novel regeneration technologies based on the promotion of neuronal migration.

## Methods

### Animals

All animal experiments were performed in accordance with the guidelines and regulations of Nagoya City University (animal protocol number 21-028). Both sexes of mice were used in the experiments. Mice were housed ≤7 per cage under the condition of $23 \pm 1\,°C$ and humidity 50-70% on a 12-h light/dark cycle with *ad libitum* access to food and water. Wild-type C57BL/6 J and Institute for Cancer Research (ICR) mice were purchased from SLC (Japan). *Dcx-EGFP* mice[67] (substrain; ICR) were provided by the Mutant Mouse Research Resource Center (MMRRC_000244-MU). *NSE-DTA* mice[49,50] (substrain; C57BL/6 J) were provided by Dr. Shigeyoshi Itohara (RIKEN, Japan).

### Plasmids

pCAGGS-EmGFP was employed[20]. pCDNA3.1-EB3-EGFP was kindly provided by Dr. Kozo Kaibuchi (Fujita Health University). pCS2-EGFP-UtrCH was kindly provided by Dr. William Bement (University of Wisconsin-Madison) and Dr. David J. Solecki (St. Jude Children's Research Hospital)[68]. pAcGFP1-actin was purchased from Clontech (accession no. 632453). Venus-CAAX and tdTomato-CAAX fragments were amplified by PCR and inserted into pENTR-D-TOPO vectors (Invitrogen). The gateway system (Invitrogen) was used to generate pCAGGS-Venus-CAAX and pCAGGS-tdTomato-CAAX expression vectors. For the construction of flag-Cortactin plasmid, Cortactin cDNA (accession no. BC011434) was amplified by PCR from the mouse V-SVZ cDNA library and inserted into pENTR-D-TOPO vector (Invitrogen). KD-resistant and Y421A-mutated Cortactin plasmids were generated by inverse PCR with modified primers (KD-resistant plasmid, miR-target sequence: 5′- CAG GAT GAT GGA GGA GCT GAT -3′, KD-resistant mutation: 5′- CAA GAC GAT GGA GGT GCA GAT -3′; Y421A-mutated plasmid, Y421: TAT, Y421A: GCT) and subsequent self-ligation. PTPσ-KD, Liprin-α-KD, and Cortactin-KD plasmids were generated as the following. Briefly, the target sequences of mouse *ptprs* [https://www.ncbi.nlm.nih.gov/nuccore/BC052462], *ppfia1* [https://www.ncbi.nlm.nih.gov/nuccore/BC157936], *stx7* [https://www.ncbi.nlm.nih.gov/nuccore/BC132123], *dstn* [https://www.ncbi.nlm.nih.gov/nuccore/BC131926], *cttn* [https://www.ncbi.nlm.nih.gov/nuccore/BC011434.1], and *lacZ* (control) mRNA were inserted into a BfuAI site of a modified Block-iT Pol II miR RNAi expression vector containing DsRed-express or EmGFP. The sequences of oligonucleotides are listed in Supplementary Data 2. Invitrogen's Gateway system was used to generate microRNA (miR) plasmids [pCAGGS-DsRed-express-miR-*ptprs* (PTPσ-KD), pCAGGS-DsRed-express-miR-*lacZ* (Control), pCAGGS-EmGFP-miR-*ppfia1* (Liprin-α-KD), pCAGGS-EmGFP-miR-*stx7* (Syntaxin-7-KD), pCAGGS-EmGFP-miR-*dstn* (Destrin-KD), pCAGGS-EmGFP-miR-*cttn* (Cortactin-KD), pCAGGS-EmGFP-miR-*lacZ* (Control), pCAGGS-flag-Cortactin*-IRES-mCherry, and pCAGGS-flag-Cortactin*Y421A-IRES-mCherry]. pCAGGS-EmGFP-miR-*fyn* (Fyn-KD) was employed[69]. The plasmid of Myc-DDK-tagged mouse ORF clones were purchased from OriGene [Syntaxin-7 (Cat. MR221576) and Destrin (Cat. MR222274)]. Vector KD efficiency was assessed by western blotting and immunocytochemistry.

### Primary neuronal culture

Cortical tissues dissected from embryonic day 15.5 – 16.5 (E15.5-16.5) embryos were dissociated with 0.05% trypsin-EDTA (Invitrogen) in L15 medium (Invitrogen). The cells were further dissociated with dissociation buffer (bovine serum albumin 3 mg/ml, MgSO₄· 7H₂O 3 mg/mL, DNase I 40 μg/ml, trypsin inhibitor from soybean 0.4 mg/ml, 10 mM HEPES in Hank's balanced salt solution (HBSS) and transfected with 1.8 μg plasmid DNA with Amaxa Nucleofector II (Lonza; Program O-005). The transfected cells were recovered at $37\,°C$ for 15 min in RPMI-1640 medium, allowed to aggregate, and then embedded in 70% Matrigel (BD Biosciences) in L15 medium. Cell aggregates were cultured in Dulbecco's Modified Eagle's Medium (Sigma Aldrich) containing 10% fetal calf serum, 2 mM L-glutamine, 2% NeuroBrew-21 (Invitrogen), and 50 U/ml penicillin-streptomycin (Gibco) for 48–72 h before fixation and 120 h before live imaging.

Primary culture of embryonic and neonatal migrating neurons was performed as the following. V-SVZ tissues dissected from P0-1 mice and lateral and medial ganglionic eminences dissected from E13.5 embryos were dissociated with 0.25% trypsin-EDTA (Invitrogen). The cells were washed in L15 medium (Invitrogen) containing 40 μg/ml DNase I (Roche) and transfected with 1.8 μg plasmid DNA with a 4D-Nucleofector X unit (Lonza, Program CL-133). The transfected cells were recovered at room temperature (RT) for 10 min, resuspended in RPMI-1640 medium, allowed to aggregate, and then embedded in 60% Matrigel (BD Biosciences) in L15 medium with or without 30 μg/ml recombinant mouse Neurocan (R&D Systems, 5800-NC-050). The cell aggregates were cultured in Neurobasal medium containing 2% NeuroBrew-21 (Invitrogen), 2 mM L-Glutamine (Gibco), and 50 U/ml penicillin-streptomycin (Gibco) for 48–72 h.

Stripe assays were performed as described previously[20]. Briefly, 10 μM bovine serum albumin-alexa647, with or without 2 mM recombinant mouse Sdc2 (R&D Systems, 6585-SD-050), in HBSS solution was injected into silicon matrices (40-μm; from Martin Bastmeyer, Karlsruhe Institute of Technology)[39] on glass-bottom dishes (Matsunami). After a 30-min incubation at $37\,°C$, the dishes and matrices were washed with HBSS, and the matrices were removed from the dishes carefully. Dishes with Sdc2 stripes were dried at RT for 10 min before being used for migrating neuron assays.

### Immunocytochemical screening of growth cone molecules

Primary cultures of migrating neurons dissected from the V-SVZ tissues of P0–1 Dcx-EGFP mice were fixed with warm 4% paraformaldehyde (PFA) in 0.1 M phosphate buffer (PB; pH 7.4) at $37\,°C$ for 15 min. Cells were incubated at room temperature (RT) for 30 min in blocking solution [10% normal donkey serum and 0.2% TritonX-100 in phosphate-buffered saline (PBS)]. Candidate molecules that localized in axonal growth cones were selected from our previous studies[31-35], incubated overnight at $4\,°C$ with primary antibodies, and incubated with AlexaFluor-conjugated secondary antibodies (1:1,000, Invitrogen) in blocking solution for 3 h at RT. Labeled cells were observed with an LSM880 confocal laser-scanning microscope (Carl Zeiss, Germany) in super-resolution mode fitted with a 40× water-immersion objective lens (NA 0.8). Image acquisition resolutions were: pixel dwell, 3.52 μs/pixel; scaling X/Y, 0.050/0.050 μm/pixel (2,380 × 600 pixels); average, 1; zoom, 1.8. The following primary antibodies were used (dilution 1:100): rabbit anti-CaMKV[34] (polyclonal, self-made), rabbit anti-LETM1[34] (polyclonal, self-made), rabbit anti-PICALM[34] (polyclonal, self-made), rabbit anti-Fish[34] (polyclonal, self-made), rabbit anti-Rab35[34] (polyclonal, self-made), rabbit anti-Arhgdia[34] (polyclonal, self-made), rabbit anti-MARCKSL1[34] (polyclonal, self-made), rabbit anti-Tmod2[34] (polyclonal, self-made), rabbit anti-pGAP43 (S96)[31] (polyclonal, self-made), rabbit anti-pRtn1[35] (polyclonal, self-made), rabbit anti-SCG10[32] (polyclonal, self-made), rabbit anti-pMAP1B (S25)[33] (polyclonal, self-made), mouse anti-Ap2a1 (610501, Becton Dickinson), mouse anti-Ap2b1 (610381, Becton Dickinson), rabbit anti-Arhgdia (sc-360, Santa Cruz), mouse anti-Calnexin (610523, Becton Dickinson), rabbit anti-Camk2 (MA1-047, Thermo Fisher Scientific), rabbit anti-Cofilin (C8736, Sigma-Aldrich), rabbit anti-Destrin (D8815, Sigma-Aldrich), mouse anti-PP2A (601555, Becton Dickinson), chicken anti-Liprin-α (PPFIA1) (GW21470, Sigma-Aldrich), chicken anti-PTPσ (PTPRS) (GW21486, Sigma-Aldrich), mouse anti-Rtn4 (612238, Becton Dickinson), rabbit anti-Scamp (121002, Synaptic Systems), rabbit anti-Snap29 (111303, Synaptic

Systems), rabbit anti-Syntaxin7 (110073, Synaptic Systems), mouse anti-Syntaxin8 (611352, Becton Dickinson), and mouse anti-Vcp (MA3-004, Thermo Fisher Scientific).

## Western blotting

To confirm the KD efficiency and resistance, plasmids expressing cDNA and miRNA were co-transfected into HEK293T cells. The cells were collected 60 h after transfection and lysed with lysis buffer (50 mM Tris-HCl pH 8.0, 100 mM NaCl, 1 mM EDTA, 1% NP-40, 0.01% sodium dodecyl sulfate [SDS]) using ultrasound sonication. The proteins were separated by SDS-polyacrylamide gel electrophoresis and transferred to PVDF membranes (Millipore), which were blocked in 1% skim milk in Tris-buffered saline containing 0.1% Tween 20 (TBS-tween). Then membranes were incubated with primary antibodies at 4 °C overnight and horseradish peroxidase-conjugated secondary antibodies at RT for 1 h. Signals were detected with enhanced luminal-based chemiluminescent western blotting reagent (GE Healthcare) using a cooled CCD camera (LAS 3000mini, Fujifilm). The following primary antibodies were used: rabbit anti-DYKDDDK antibody (1:1000, 2368, Cell Signaling Technology), goat anti-PTPσ antibody (1:1000, AF3430, R&D systems), and mouse monoclonal anti-actin antibody (1:10,000, MAB1501, Millipore).

## Image processing and quantification using Weka Segmentation

Growth cone areas of Venus-CAAX-expressing extending axons and migrating neurons in time-lapse imaging were obtained from super-resolution imaging data with the Trainable Weka Segmentation plugin in FIJI (Supplementary Fig. 1a–f). Sixty maximum-intensity projection images from six cells (10 images/cell) were used as a training dataset to prepare an algorithm for classifying growth cone areas and other areas. For protein quantification within growth cones of migrating neurons during screening, images of ten randomly selected cells with migratory bipolar morphology were acquired. We used the Trainable Weka Segmentation plugin in FIJI to establish an algorithm for axonal growth cone segmentation in the fixed samples. Forty images of Dcx-EGFP+ axons were used as a training data set to establish the algorithm. The Gaussian blur, Hessian, Membrane projections, Sobel filter, Difference of Gaussians, Structure, and Neighbors segmentation settings in the plugin were selected for machine learning; and then area quantification in the LPs was performed with the Analyze Particles tool in FIJI. In the screening of growth cone molecules, the intensity ratio was calculated as follows: mean signal intensity in the identified growth cone / mean signal intensity in the leading shaft. Cytosolic EGFP signal was used for control (Supplementary Fig. 2e) and compared with growth cone molecule data by one-way ANOVA followed by the Dunnett test.

## Immunocytochemistry on cultured neurons

For immunocytochemistry of axonal and LP growth cones, primary neuronal culture was fixed with warm 4% PFA in 0.1 M PB at 37 °C for 15 min. Src-family tyrosine kinase inhibitor PP2 (20 μM) was added into cultured medium 15 min before fixation. The following primary antibodies were used: mouse monoclonal anti-acetylated tubulin (1:200, T6793, Sigma-Aldrich); mouse monoclonal anti-Cortactin (1:100, 05-180, Millipore); rabbit anti-pY421-Cortactin (1:100, 4569 S, Cell Signaling Technology); rabbit anti-pY421-Cortactin (1:100, AB3852, Sigma-Aldrich); rabbit anti-Destrin (1:100, D8815, Sigma-Aldrich); rabbit anti-Syntaxin-7 (1:200, 110073, Synaptic Systems); rat anti-tyrosinated tubulin (1:200, MAB1864-I, Sigma-Aldrich); rabbit anti-PPFIA1 (1:100, HPA042271, Sigma-Aldrich); chicken anti-PPFIA1 (1:100, GW21470, Sigma-Aldrich); chicken anti-PTPRS (1:100, GW21486, Sigma-Aldrich); rabbit anti-GFP (1:1,000, 598, MBL life science); rat anti-GFP (1:1,000, 04404-84, Nacalai); rabbit anti-DsRed (1:1000, 632496, Clontech), and rabbit anti-DCX (1:500, 4604, Cell Signaling Technologies). F-actin and nuclei were visualized with Alexa Fluor 488

Phalloidin (1:100, A12379, Invitrogen) and Hoechst 33342 (1:5,000, Invitrogen), respectively.

Image stacks of fluorescence-labeled cultured migrating neurons and elongating axons were acquired with an LSM880 confocal laser-scanning microscope equipped with Airyscan FAST (Carl Zeiss, Germany) in super-resolution mode with a 40× water-immersion (NA 1.2; optical zoom, 1.8×) or 63× oil-immersion (NA 1.4; optical zoom, 3.0×) objective lens. Image acquisition resolutions were: pixel dwell, 3.52 μs/pixel; scaling X/Y, 0.050/0.050 μm/pixel; average, 1; zoom, 1.8.

## Time-lapse super-resolution imaging of elongating axons and migrating neurons

In vitro time-lapse super-resolution imaging was performed as described previously[25]. Cells were cultured in a stage top chamber at 37 °C in a 5% $CO_2$ incubation system (Tokai Hit STXG-WSBX-SET). Image stacks of fluorescence-labeled migrating neurons and elongating axons were acquired with an LSM880 confocal laser-scanning microscope equipped with Airyscan FAST (Carl Zeiss, Germany) in super-resolution mode with a 40× water-immersion (NA 1.2; optical zoom, 1.8×) or 63× oil-immersion (NA 1.4; optical zoom, 3.0×) objective lens, with an interval of 30 s (40× objective lens) or 3.0–6.0 s (63× objective lens), respectively. Image acquisition resolutions were: pixel dwell, 0.74 μs/pixel; scaling X/Y, 0.050/0.050 μm/pixel; 1.0-μm z-step size for the 40× water-immersion objective lens; and pixel dwell, 0.99 μs/pixel; scaling X/Y, 0.040/0.040 μm/pixel; 0.189-μm z-step size for the 63× oil-immersion objective lens. To avoid focus drift, the Definite Focus 2 function was used in every imaging frame.

## Image processing and quantification

For analyses of super-resolution images, original Airyscan images were acquired as 16-bit data and then processed and converted into 8-bit tiff images in ZEN (Carl Zeiss, Germany)[25]. The Trainable Weka Segmentation plugin in FIJI was used to classify growth cone and other areas.

The distribution of stained PTPσ in neurons was measured in Liprin-α-KD and control neurons as follows. The intensity of PTPσ signal at the membrane of the leading shaft and soma was measured at 0.1-μm intervals. The intensity at the somal membrane was averaged and set as baseline intensity. The PTPσ-concentrated position in the leading shaft was defined as the position where signal intensity was 2.5-fold higher than the baseline intensity. When > 5% of the total length of the leading shaft was PTPσ-concentrated, the distribution of PTPσ was defined as being in both the growth cone and leading shaft, while when it was ≤ 5% of total leading shaft length, the distribution of PTPσ was defined as being only in the growth cone.

All of the Z-stacked time-lapse images of extending axons and migrating neurons (Figs. 1–3; Supplementary Fig. 1, 4, 5) were shown as maximum intensity projection images, which were processed using ZEN software (Carl Zeiss, Germany). In the CS-PTPσ functional analysis, original single z-plane images showing the border of Alexa647-labeled- and non-colored-Matrigels were shown in Supplementary Fig. 2i, and their maximum intensity projection images with Alexa647-labeled gel area (colored with semi-transparent magenta) were shown in Fig. 1k–n.

To measure speed of migrating neurons, we used the Manual Tracking plugin in FIJI software or ZEN software. Migrating neuron activities were compared between non-coated regions and coated stripes. Maximum growth cone areas were measured in ZEN software (Carl Zeiss, Germany). Growth cone filopodium formation degree and contraction in rest and extending phases were recorded every 20th frame for ≥300 s in ZEN software. The degree and frequency of formed lamellipodia in axonal and LP growth cones were recorded from at least 100 sequential imaging frames. The frequency of EB3-EGFP+ dots in filopodia was determined at rest (control) and in extending growth cones. In the photoswitchable inhibitor experiments, the intensities of EGFP-UtrCH+ signals and EB3-EGFP+ dots in the tdTomato-CAAX+ leading filopodia and leading shafts during leading filopodium

formation (2-4 min from the invasion into the Sdc2 stripe area) and retraction (4-6 min from the invasion into the Sdc2 stripe area) were measured in ZEN software. Signal intensity of axonal growth cone molecules, Cortactin, and pY421-Cortactin in the LP growth cones (Fig. 2k; Supplementary Fig. 2a-d and 3j) were measured in ZEN software and shown after subtraction of background intensity.

## HSPG screening assay
The following recombinant HSPGs, containing extracellular domain with sulfated glycosaminoglycan heparan sulfate, were purchased from R&D Systems: Gpc1 (cat. 4520-GP-050), Gpc2 (cat. 2355-GP-050), Gpc3 (cat. 6938-GP-050), Gpc4 (cat. 9195-GP-050), Gpc5 (cat. 2689-G5-050), Gpc6 (cat. 9429-GP-050), Sdc1 (cat. 3190-SD-050), Sdc2 (cat. 6585-SD-050), Sdc3 (cat. 2734-SD-050), Sdc4 (cat. 6267-SD-050). Glass-bottom dishes or glass-cover slips were coated 2 µg/ml with FITC-conjugated anti-human-IgG Fc antibody (Sigma Aldrich, cat. F9512) in HBSS (Gibco) with or without 2 mM recombinant HSPGs. Briefly, a drop of recombinant HSPGs was placed on a glass-cover slip, incubated 30 min at 37 °C, and then dried for 10 min at RT. The coating was rinsed with HBSS to remove unbound proteins and dried for 10 min before proceeding with V-SVZ cultures. Some coated Sdc2 was treated with 2 U/ml heparinase III (Sigma Aldrich, cat. H8891) for 30 min at 37 °C. The cells were cultured for 36 h and fixed with 4% PFA in 0.1 M PB (pH 7.4) for 20 min at RT, then subjected to immunocytochemistry with anti-DCX antibody and Hoechst staining as described above. Images of Dcx+ cells contacted on the dish bottom (labeled with FITC) were acquired by an LSM700 confocal laser-scanning microscope (Carl Zeiss, Germany) with a 20× dry objective lens (NA 0.8). Images were either tiled and the perimeter of the explanted V-SVZ and the total area of migrated Dcx+ cells were measured in ImageJ (version 2.3.0, NIH), or LP lengths of Dcx+ cells were measured in ZEN software (Carl Zeiss, Germany). For measurement of migration speed on HSPG-coated dish, time-lapse video recordings were captured with a BZ-X800 fluorescence microscope (Keyence, Japan) equipped with a 20× dry objective lens (NA 0.75). Images of migrating neurons were obtained every 5 min for 24 h under humidified conditions at 37 °C with 5% $CO_2$. Migration speed was analyzed with the Manual Tracking plugin in FIJI.

## Photoswitchable chemical inhibitors
Photoswitchable chemical inhibition was performed as reported previously[70] with modification. We conducted time-lapse super-resolution imaging of 5 µM Opto-Lat-, 0.4 µM PnOJ-, or 2 µM PST-1-treated cultured migrating neurons using an LSM880 confocal laser-scanning microscope with Airyscan FAST (Carl Zeiss, Germany) in super-resolution mode with a 40× water-immersion (NA 1.2; optical zoom, 1.8×) objective lens and 30-s interval. The fiber outputs of the 405-nm and Argon (500-nm) lasers in our microscope system were 15.08 mW and 12.33 mW, respectively. Opto-Lat, PnOJ, and PST-1 were photoswitched by 405-nm (2.0%) and 514-nm (2.0%) laser-illumination for 6–8 s in a squared ROI (4 µm-width) set on Sdc2 stripes. Z-axis focus was maintained with the Definite Focus 2 function and auto-adjusted by illuminating IR laser at the beginning of each imaging session to ensure precise spatial photo illumination. The imaging and illumination settings in the photo-illumination experiments were as follows: imaging scaling X/Y, 0.050/0.050 µm per pixel; z-interval, 1.0 µm; Zoom 2.0; pixel dwell, 0.98 µs; average, 1; and bleaching iteration, 20. Details on Opto-Lat, PnOJ, and PST-1 are described elsewhere[44−46]. Concentrations of photoswitchable inhibitors were determined based on the observation of inhibitory effects of actin and microtubule dynamics (Supplementary Fig. 4d–g) and a previous report[70].

## Liquid chromatography /mass spectrometry of mouse tissues for proteomics
Proteomics analysis was performed as previously reported[31,34]. Briefly, cerebral cortex and RMS tissues from P0 pups were dissected, freshly frozen in liquid nitrogen, and stored in a deep freezer. The tissues were lysed with phase-transfer surfactant buffer (12 mM sodium deoxycholate, 12 mM sodium lauroyl sarcocinate, 50 mM ammonium bicarbonate, cOmplete protease inhibitor cocktail (Merck) and phosphatase inhibitors (200 mM Imidazole, 100 mM NaF, 115 mM $Na_2MoO_4$, 100 mM $Na_3VO_4$). The lysates were sonicated on ice and subsequently centrifuged at 20,400x $g$ for 15 min at 4°C. The supernatants were reduced with dithiothreitol, carbamide methylated with iodoacetamide in the dark and subsequently subjected to in-solution trypsin digestion with 2 mM $CaCl_2$ for 18 h. Trypsin solution was added twice at enzyme/substrate ratio of 1: 50, the second addition was performed 3 h after the first addition. After enzyme digestion, the detergents were removed by liquid-liquid extraction using ethyl acetate with 1.0 % trifluoroacetic acid (TFA) and the aqueous phases were dried under vacuum. The peptide samples dissolved in 0.2% TFA were desalted with Monospin C18 columns (GL Sciences) and highly hydrophilic peptides, still remaining in the flow-through from Monospin C18 columns, were collected using InterSep GC (GL Sciences). The eluted peptide solutions were dried under vacuum again and stored at −80°C until MS analysis. The shotgun proteomics method used herein has been described previously[31,34]. The dried peptides were finally dissolved in 0.3% formic acid and injected into a nano-flow liquid chromatography system (ekspert nanoLC 415 system; Eksigent Technologies) coupled with a tandem MS (TripleTOF 5600; Sciex). Sample analysis was conducted in triplicate under direct injection mode with an analytical column (75 µm × 150 mm, 3 µm particle diameter; Nikkyo Technos). Mobile phases A and B were 0.1% FA and 0.1% FA in acetonitrile, respectively. Peptides were eluted using a 40 min gradient from 2 to 32% B at 300 nl/min. MS spectra (250 ms), followed by 10 MS/MS spectra (100 ms each), were acquired in data-dependent mode. Protein identification was carried out using the Mascot version 2.6.0 (Matrix Science) search engine. The proteome data generated in this study have been deposited in the PRIDE database under accession code PXD048878.

The proportion of total cells that were Dcx+ in the RMS tissues was $81.6 \pm 6.9$ % ($n = 3$ pups). Growth cone-related proteins were extracted with the following parameter settings: (1) normalized spectral abundance factor (NSAF): 0.8 <cortex/RMS ratio <1.25 (Supplementary Data 1); (2) gene ontology (cellular component): growth cone; (3) gene ontology (biological process): neuronal projection development. Gene ontology analysis was performed using the PANTHER classification system (pantherdb.org).

## Preparation of biomaterials
GFs were prepared with a manufacture modification on Genocel-L[71]. Briefly, gelatin powder (isoelectric point 5.0, viscosity 4.6 mPas and gelatin bloom strength 250 g; Nitta Gelatin, Inc., Japan) was immersed in distilled water and dissolved completely to obtain 37.5 w/v% gelatin solution. The solution was heated to 60 °C and ejected through a 250-µm wide nozzle. The gelatin product was air-dried with heat until it had solidified and then subjected to dehydrothermal crosslinking at 140 °C for 12 h or at 160 °C for 24 h. Crosslinked GFs were die-cut into 5 mm × 5 mm sheets. Gelatin sponge was prepared from 5 w/v% gelatin solution. Drops of heated gelatin solution were allowed to gelatinize at RT and then subjected to lyophilization for 60 h and brief heat-crosslinking under pressure ($1 \times 10^{-5}$ MPa) at 160 °C for 24 h. PFs were prepared with the same diameter and structure as swollen GFs[47].

## X-ray computed tomography (CT)
Dried GF sheets were immersed in PBS and subsequently fixed with 4% PFA overnight at RT. Fixed fabrics were rinsed with PBS, dehydrated in a graded ethanol series (70%, 80%, 90%, and 100%), and stained with 1% iodine in 100% ethanol for 3 h. Stained fabrics were washed 3× in 100% ethanol and cut into small pieces. Gelatin fabrics were examined with a laboratory-based X-ray CT Rigaku nano3DX (Rigaku, Japan), equipped with

with a copper X-ray target (tube settings: voltage, 40 kV; current, 30 mA). Gelatin fiber structure was analyzed at a voxel size of 0.6 μm. The obtained CT data were processed using the visualization software package Avizo (Thermo Fisher Scientific). The averaged orientation of individual fibers was measured from the orientation tensors[72] using the XFiber extension plugin in Avizo. The voids among fibers were extracted by computing the ambient occlusion module scalar field in a labeled field (https://diglib.eg.org/handle/10.2312/eurovisshort20161171). The size of the void was evaluated using the thickness-map module[73], which computes a voxel-wise thickness for three-dimensional objects.

## Scanning electron microscopy

The intrastructures of dried biomaterials were observed using a scanning electron microscope (FlexSEM 1000, Hitachi High Technologies Corp., Japan). The samples were placed on the microscope stage, fixed with conductive tape, and then coated with sputtered gold using a sputter coater (MSP-1S, Vacuum Device Inc., Japan). The secondary electron images were obtained under the low vacuum mode (30 Pa) at 5 kV acceleration voltage.

## Cryogenic injury

Cortical cryogenic injury was performed on P2 mice as described previously[20,64,65]. Briefly, mice were anesthetized with 1% isoflurane in oxygen (300 ml/min), affixed to stereotaxic instruments (David Kopf Instruments, USA), and their skulls were exposed by a scalp incision. A chilled 1.2-mm-wide hexagonal wrench was prepared in liquid nitrogen. The metal probe was placed on the exposed right skull (0.5-mm anterior and 1.2-mm lateral to lambda) for 10 s, three times, with a 5-s time interval during which the probe was re-chilled. The scalp was sutured, Mice were placed in a heated recovery box for 15 min, and then returned to their home cages.

## Biomaterial implantation in the injured cortex

Dried GFs (5 mm × 5 mm), gelatin sponge or PFs were incubated in augmenting solution overnight at 4 °C prior to implantation into cryoinjured sites. After being immersed in solution, GF fiber thickness had increased whereas PF fiber thickness had not changed. P5 mice were deeply anesthetized by spontaneous inhalation of 1–3% isoflurane. The scalp and parietal bone at the cryoinjured site were resected to expose the brain. The biomaterials were cut into 1 mm × 1 mm pieces and placed in injured sites with tweezers. The scalp was sutured.

## Immunohistochemistry

Animals were deeply anesthetized with isoflurane and perfused transcardially with PBS, followed by 4% PFA in 0.1 M PB. Brain sections were processed using a vibratome VT1200S (Leica, Germany). 50- and 60-μm-thick coronal sections were prepared from P30 and P9 fixed brains, respectively. The sections were incubated in blocking solution (10% normal donkey serum, 0.4% Triton X-100 in PBS) for 1 h at RT, then in primary antibody solution overnight at 4 °C, followed by secondary antibody solution containing AlexaFluor-conjugated secondary antibodies (1:1,000, Thermo Fisher Scientific) for 2 h at RT. Signals were amplified with biotinylated secondary antibodies (1:500, Jackson Laboratory) in conjunction with a Vectastain Elite ABC kit (Vector Laboratories), and visualized with the TSA Fluorescence System (Akoya Biosciences). The following primary antibodies were used: rabbit anti-DCX (1:500, Cell Signaling Technologies), mouse anti-CS56 (1:500, Sigma Aldrich), chicken anti-PTPRS (1:200, Sigma Aldrich), rat anti-GFP (1:500, Nacalai), and rabbit anti-NeuN (1:200, Abcam). Cell nuclei were stained with Hoechst 33342 (1:5,000, Invitrogen). Some sections were treated with 0.2 U/ml chondroitinase ABC (Sigma Aldrich, C3667-5UN) for 2 h at 37 °C prior to CS56 antibody incubation.

Coronal sections including the V-SVZ were used for quantitative analysis. Confocal z-stack images of immunolabeled areas were captured with an LSM700 confocal laser microscope (Carl Zeiss, Germany) equipped with 20× dry (NA 0.8) and 40× water-immersion (NA 1.2) objective lenses, or an Olympus FV3000 confocal microscope (Evident, Japan) equipped with 20× (NA 0.8) objective lens. ZEN software (Carl Zeiss, Germany) was used for the CS-staining intensity and cell density measurement. Image stacks of growth cones from Dcx-EGFP+ migrating neurons in injured cortices implanted with HSPG-enriched GFs were acquired, using Nikon Eclipse Ti2-E confocal laser-scanning microscope equipped with AX R with NSPARC (Nikon, Japan) in super-resolution mode with a 40× water-immersion (NA 1.15; optical zoom, 5.8×) objective lens with 0.5-μm z-step size. Image acquisition resolution was 0.074 μs/pixel (Supplementary Fig. 7a). The number of EmGFP+NeuN+ co-labelled cells in every second section containing the lesion track was counted and the cell count number was multiplied by two to obtain the estimated number of total cells per injured cortex sample.

## Serial block-face scanning electron microscopy

Serial block-face scanning electron microscopy was performed as described previously[25,70] with slight modifications. P9 *Dcx-EGFP* mouse brains were fixed by transcardial perfusion with 2.5% glutaraldehyde and 2% PFA in 0.1 M PB (pH 7.4) at 4 °C, and postfixed in the same fixative for 2 d at 4 °C. Coronal sections (150-μm-thick) were prepared with a vibratome (VT-1200S, Leica), and a part of the injured cerebral cortex (1 mm × 1 mm) was dissected from these sections using an ophthalmic knife under a stereomicroscope. These blocks were treated with 2% OsO4 and 1.5% potassium ferricyanide in PBS for 1 h at 4 °C, 1% thiocarbohydrazide for 20 min at RT, 2% aqueous OsO4 for 30 min at RT, uranyl acetate solution overnight at 4 °C, and lead aspartate solution for 2 h at 50 °C. The samples were then dehydrated in a graded ethanol series (60%, 80%, 90%, 95%, and 100%) and treated with dehydrated acetone. Coronal cortical sections (1 mm × 1 mm × 150 μm) were embedded in Durcpan resin containing 8% Ketjen black powder and put on the rivets for two overnights at 60 °C to ensure polymerization. Images of cortical tissues were acquired by a Merlin and Sigma scanning electron microscope (Carl Zeiss, Germany) equipped with a 3View® in-chamber ultramicrotome system (Gatan). Serial image sequences were 65.14 μm × 42.60 μm wide (8.0 nm/pixel) and >120 μm deep at 80 nm steps. Sequential images were processed in FIJI software (version 2.0.0). Migrating neurons were identified based on reported ultrastructural features[25]: a smooth cell contour, dark cytoplasm with a small Golgi apparatus, short endoplasmic-reticulum fragments, and extracellular spaces. Segmentation of cell contours was performed in Microscopy Image Browser software and 3D reconstructions were performed in Amira software (Maxnet Co., Ltd., Japan). Components of three-dimensional object files (.obj) obtained from Amira software were combined by Blender software (https://www.blender.org/) and visualized as interactive 3D models by Sketchfab (https://sketchfab.com).

## Transmission electron microscopy

*Dcx-EGFP* mice subjected to cortical injury at P2 and recombinant HSPGs or vehicle-treated GF transplantation at P5 were fixed at P9 by transcardial perfusion with 2.5% glutaraldehyde and 2% PFA in 0.1 M PB (pH 7.4) at 4 °C, and postfixed in the same fixative for 2 d at 4 °C. Coronal sections (200-μm-thick) were prepared with a vibratome (VT-1200S, Leica). Sections that covered the center of injury, the grafted GF pieces, and GFP-labeled new neurons in the cortex were selected, treated with 4% OsO4 in 0.1 M PB for 90 min, dehydrated in graded ethanols and propylene oxide, embedded in epoxy resin (Durcupan, Sigma Aldrich), and heated in an oven at 60 °C for 72 h to induce resin polymerization. Serial semi-thin sections (1.5-μm-thick) were cut with an ultramicrotome (UC7, Leica) using a diamond knife (Histo, Diatome, USA), and stained lightly with 1% toluidine blue. After image acquisition with a transmitted light microscope (Olympus BX51, Evident, Japan),

semi-thin sections were glued to Durcupan blocks and detached from the glass slide by repeated freezing in liquid nitrogen and thawing. Ultra-thin sections (60–70-nm-thick) were prepared from the semi-thin sections with an ultramicrotome (UC7, Leica) and a diamond knife (SYM2045, Syntek, Japan), stained with 2% uranyl acetate for 15 min, and stained with modified Sato's lead solution (Reynolds' solution) for 5 min. Images were captured by a transmission electron microscope (JEM-1400Plus, JEOL, Japan) equipped with a digital camera.

Under transmission electron microscopy, gelatin fibers could be recognized as electron-dense organelle-free thick fibers filled with a meshwork of small fibrous structures in necrotic tissues where activated microglia containing many vacuoles and lysosomes had accumulated. Migrating neurons were identified by their smooth and elongated cell body, a small elongated nucleus with lax chromatin, and scant cytoplasm containing many free ribosomes and microtubules[21,25]. Astrocytes were recognized by their electron-lucent cytoplasm containing many intermediate filaments and glycogen granules[21].

### Time-lapse imaging of migrating neurons in brain slices

*Dcx-EGFP* mice were cryoinjured at P2 and implanted with GF swollen with DyeLight650 (1:100, 84535, Invitrogen) solution with or without HSPGs (2 mM) at P5. Dissected P7-8 cryoinjured brains were sliced into 220-μm thick sections with a vibratome (VT1200S, Leica, Germany). Brain tissue including the biomaterial was cut into 250-μm thick slices with Tissue Chopper (McIlwain TC752, Campden Instruments, US). The slices were placed on a stage-top imaging chamber (Warner instruments, US) under continuous perfusion of artificial cerebrospinal fluid (37 °C, bubbled with 95% $O_2$ and 5% $CO_2$) during imaging. Time-lapse video recordings were obtained via a FV3000 confocal microscope (Evident, Japan) equipped with 20× dry (NA 0.8) and 40× oil-immersion (NA 1.2) objective lenses. Every 3–8 min, 25–40 z-sections (1.2–1.8 μm steps) images were obtained automatically over 8–12 h. Focal planes were merged to visualize entire cells. Migration speed was measured by tracking cells in the vicinity of gelatin fibers with the FIJI Manual Tracking plugin.

### V-SVZ electroporation

Neonatal pups (P0–1) were anesthetized by brief hypothermia and fixed on a stereotaxic instrument. A pulled-out glass capillary containing 2 μl of plasmid solution (4 μg/μl pCAGGS-EmGFP in distilled water containing 0.01% Fast Green) was placed into the right lateral ventricle (2.0 mm anterior, 1.2 mm lateral to lambda and 2.0 mm deep) and plasmid was injected. The animals were subjected to three electrical pulses (70 V, 30.0 ms) via a Super Electroplater NEPA21 Type II (Nepa Gene, Japan) and 10-mm tweezer electrodes (CUY650P10, Nepa Gene, Japan). Electroplated animals were placed on a heating plate before being returned to their home cages.

### Adenoviral transduction

Adenovirus encoding Cre recombinase under the control of the cytomegalovirus promoter Ad-iCre-GFP was purchased from Vector Biolabs (Cat. 1772). Newborn *NSE-DTA* mice or C57BL/6 J mice were anesthetized by brief hypothermia and fixed on the stereotaxic instrument. A glass capillary was filled with 1 μl viral solution, placed into the right lateral ventricle (2.0 mm anterior, 1.2 mm lateral to lambda, and 2.0 mm deep), and the virus was injected.

### Foot-fault test

The foot-fault test was performed on P29 cryoinjured mice as previously described[20,64]. Briefly, mice roamed on 20-cm elevated hexagonal wire grids with 40-mm diameter openings for 10 min 1 d prior to the test. For the test, each mouse was placed on the grid for 5 min and the total number of foot-faults for each hindlimb was recorded. A misplaced limb that slipped on the grid or fell through the openings in the grid was counted as a foot-fault. The number of foot-faults for the left hindlimb

was divided by the total number of hindlimb foot-faults to determine the fault rate. The test was repeated twice on the same day for each mouse and the percentage of left foot-faults was averaged.

### Statistics and Reproducibility

All of the data were two-tailed, and sample sizes were chosen based on previous studies. Data distributions were analyzed with Kolmogorov-Smirnov or Shapiro-Wilk tests. Inter-group equality of variance in normally distributed data was analyzed with $F$ tests. Comparisons between two groups were made with unpaired $t$-tests, Welch's $t$-tests, Mann-Whitney U-test, or Fischer's exact tests. Comparisons among multiple groups were analyzed with one-way ANOVAs, or Kruskal-Wallis tests followed by a post-hoc Steel Dwass, Tukey, Dunnett, Dunn's, or Bonferroni tests or with the two-stage step-up method of Benjamini, Krieger and Yekutieli. The data are presented as the means ± standard errors (SEMs) unless described otherwise. $P$ values < 0.05 were considered significant. Unless mentioned otherwise, the experiments were at least three times repeated independently with similar results. Statistical data, biological replicate numbers, and individual data points are listed in the Source data file.

### Reporting summary

Further information on research design is available in the Nature Portfolio Reporting Summary linked to this article.

## Data availability

The mass spectrometry proteomics data generated in this study have been deposited to the ProteomeXchange Consortium via the PRIDE partner repository under accession code PXD048878. Additional data supporting the findings are provided in the Supplementary Information. Source data are provided with this paper.

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

## Acknowledgements

We are grateful to S. Itohara (RIKEN, Japan), K. Kaibuchi (Fujita Health University, Japan), H. Nakagawa (Nagoya City University, Japan), W. Bement (University of Wisconsin-Madison), and D. J. Solecki (St. Jude Children's Research Hospital) for materials; A. Kawasaki and M. Nozumi (Niigata University, Japan), S. Nakamura, M. Matsumoto, and H. Takase (Nagoya City University, Japan), M. Furuse and A. Imai (National Institute of Physiological Sciences, Japan) for technical supports; the Research Equipment Sharing Center and the Center for Experimental Animal Science at the Nagoya City University for providing technical and animal supports; and Sawamoto laboratory members for discussions. This work was supported by research grants from Japan Agency for Medical Research and Development (AMED) (23gm1210007, 21bm0704033h0003 [to K.S.], 21jm0210060 [to N.K., K.S., and M.S.]), Japan Society for the Promotion of Science (JSPS) KAKENHI (26250019, 17H01392, 19H04785, 18KK0213, 20H05700, JP22H04926 [to K.S.], 21K06395 [to M.S.], 23K19406 [to C.N.]) and Core-to-Core Program (JPJSCCA20230007 [to K.S.]), Bilateral Open Partnership Joint Research Projects (to K.S.), Grant-in-Aid for Research at Nagoya City University (to K.S. and M.S.), the Spanish Ministry of Science, Innovation and Universities (PCI2018-093062 [to V.H.-P.]), the Valencian Council for Innovation, Universities, Science and Digital Society (Prometeo/2019/075 [to J.M.G-V.]), the Deutsche Forschungsgemeinschaft (Cluster of Excellence EXC2051 Balance of the Microverse— Microverse—Project-ID 390713860, equipment grant INST 275/442-1 FUGG (to H.-D.A.), Deutscher Akademischer Austauschdienst (to V.N. and H.-D.A.), Studienstiftung des deutschen Volkes (to N.A.V.), Cooperative Study Programs of National Institute for Physiological Sciences (to K.S.), the Mitsubishi Foundation (to K.S.), the Canon Foundation (to K.S.), the Nitto Foundation (to M.S.), the Hori Science & Arts Foundation (to M.S.), Terumo Life Science Foundation (M.S.), and the Takeda Science Foundation (to K.S. and M.S.).

## Author contributions

C.N., M.S., E.U., Y.Takagi, N.N., K.K., N.K., S.Y., and H.N. performed experiments. N.S., K.N., K.M., and S.U. produced biomaterial. H.U., M.N., and Y.O performed X-ray CT experiments. N.A.V., F.K., V.N., H.-D.A., and D.T. produced and characterized photoswitchable inhibitors. Y.I. acquired and analyzed proteomics data. V.H-P. and J.M.G-V. analyzed the TEM data. N.O. analyzed SBF-SEM data. M.I. produced antibodies. K.S. conceived the project. The work was supervised by Y.Tabata, M.I., and K.S. The manuscript was written by C.N., M.S., and K.S., with contributions from all authors.

## Competing interests

The authors declare no competing interests.

## Additional information

[1]Department of Developmental and Regenerative Neurobiology, Institute of Brain Science, Nagoya City University Graduate School of Medical Sciences, Nagoya 467-8601, Japan. [2]Division of Neural Development and Regeneration, National Institute for Physiological Sciences, Okazaki 444-8585, Japan. [3]Laboratory of Neuronal Regeneration, Graduate School of Brain Science, Doshisha University, Kyoto 610-0394, Japan. [4]Research and Development Center, The Japan Wool Textile Co., Ltd., Kobe 675-0053, Japan. [5]Medical Device Department, Nikke Medical Co., Ltd., Osaka 541-0048, Japan. [6]Laboratory of Biomaterials, Department of Regeneration Science and Engineering, Institute for Life and Medical Sciences (LiMe), Kyoto University, Kyoto 606-8507, Japan. [7]Department of Chemistry, New York University, New York, NY 10003, USA. [8]Institute for Organic Chemistry and Macromolecular Chemistry, Friedrich Schiller University Jena, Jena 07743, Germany. [9]Toray Research Center, Inc., Otsu 520-8567, Japan. [10]Department of Neurochemistry and Molecular Cell Biology, School of Medicine and Graduate School of Medical/Dental Sciences, Niigata University, Niigata 951-8510, Japan. [11]Laboratory of Comparative Neurobiology, Cavanilles Institute, University of Valencia, CIBERNED, Valencia 46980, Spain. [12]Department of Anatomy, Division of Histology and Cell Biology, Jichi Medical University, School of Medicine, Shimotsuke 329-0498, Japan. [13]Division of Ultrastructural Research, National Institute for Physiological Sciences, Okazaki 444-8585, Japan. [14]Department of Systems Pharmacology and Translational Therapeutics, Perelman School of Medicine, University of Pennsylvania, Philadelphia, PA 19104, USA. [15]These authors contributed equally: Chikako Nakajima, Masato Sawada. ✉e-mail: sawamoto@med.nagoya-cu.ac.jp

