## [Peer Review File · Nature Communications]

REVIEWER COMMENTS

Reviewer #1 (Remarks to the Author):

In this manuscript, Nakajima et al. present diverse sets of data showing the role of a growth-cone-like structure at the leading process of a migrating neuron (LP growth cone), particularly of its leading filopodium as an important structure that translates environmental molecular information into translocation of the neuronal soma. Furthermore, their research on LP growth cones has contributed to the development of a new type of biomaterial containing heparan sulfate proteoglycans (HSPGs) that facilitates neuronal migration and functional recovery after brain cortical injury. The experiments in this paper have been carefully designed, and its main conclusions are well supported by the experimental data. I think this work contributes substantially to our better understanding of cellular mechanisms underlying neuronal migration in the injured brain. I only have a few minor comments as follows:

1. Probably the most apparent difference between axonal growth and LP growth is that the latter must be accompanied by somal translocation. As explained in page 4 and shown in Figure 1b, migrating neurons exhibit alternating LP growth and pause phases with somal pausing and translocation. It seems to me that the soma moves forward when its LP is in the pause phase. Is this statistically true? If so, it would make for an interesting discussion on the presence of intracellular signaling that mediates the synchronized behavior of the LP growth cone and the soma.
2. In the first paragraph in page 6, I assume that the authors intend to compare neuronal behavior on HSPG versus control stripes. However, it is unclear which descriptions are about neurons on HSPG stripes.
3. It is stated in page 10, "Our photoinactivation experiments suggested that the proper balance of F-actin and microtubule assembly dynamics is essential for leading filopodium elongation." However, I am not sure what exactly this "balance" means. Also, I cannot find any experiments that examined the balance of cytoskeletal dynamics.

Reviewer #2 (Remarks to the Author):

In this manuscript, Nakajima and co-authors assessed the cytoskeletal dynamics of growth cones of migrating cells and their responsiveness to chondroitin sulfate (CS) proteoglycans through protein tyrosine phosphatase receptor type sigma (PTP σ). The authors further studied whether heparan sulfate (HS)-containing biomaterials implanted into the injured cortex attenuate the inhibitory effects of CS and promote neuronal migration and recovery. The authors used compelling experimental approaches, and their findings are interesting. Specifically, using super-resolution and time-lapse imaging approaches, the authors showed that growth cone of migrating cells display structural and cytoskeletal dynamics which is similar to that of axonal growth cone. They also showed that growth cone responsiveness to some micro-environmental cues depends on the same molecular pathways as in the axonal growth cones and that modulation of these pathways either ontogenetically or through HS-containing gelatin fibers affects the efficiency of neuronal migration in vitro, as well as in vivo, in the injured cortex, to promote the functional recovery.

Together these results provided a detailed characterization of growth cone cytoskeletal dynamics and revealed some of CS- and HS-activated molecular pathways. This is an interesting and comprehensive study that may be strengthened by addressing following points.

1. The authors performed a comprehensive analysis (Fig. 1; Supp Fig 1) for the expression of multiples molecular factors in the growth cone of migrating neurons. They selected factors that were reported to be enriched in the growth cone of axons. However, essentially all factors, except two, were not enriched in the growth cone of migrating neurons (Supp. Fig. 1K). The authors have used machine learning-based algorithms for the extraction of growth cones of migrating cells and calculated the intensity ratio of immunofluorescent signals between the growth cone and the shaft of the leading process. Since these approaches have not been used for axonal growth cone, it would be informative to perform similar analysis for some of these molecular cues in the axons. It will allow side-by-side comparison of molecular signatures of growth cones of axons and migrating cells, and will be in line with other comparative analysis reported by authors.

2. Some data reported by the authors are based on observations from low number of cells. In some experiments, the authors used only 3-5 cells (Figs. 1-3). The number of cells is very low and should be increased, since it may not be representative of the entire population of migrating cells.

3. Optogenetic experiments are interesting but requires additional controls with i) 405 nm illumination of the leading filopodium without photoswitchable inhibitors to show that blue light illumination alone does not change the dynamics of leading filopodium, and ii) demonstration that actin and microtubule dynamics were effectively affected by optogenetic stimulations in the leading process and shaft, in the presence of photoswitchable inhibitors. More information and validation are also required for opto-latrunculin, which is not yet published.

4. The authors quantified the percentage of migrating cells along the gelatin fibers enriched with HS. They focused their analysis at the level of cell bodies of migrating cells, which is in contrast to all previous analysis showing PTP σ enrichment in the growth cone. Can the authors look at the level of growth cone? To get better cellular resolution this can be done with electroporated mice.

5. The rational for including Gpc4 and Sdc4 in some experiments, but not in others is not clear. For example, the authors reported the distribution of Dcx-GFP+ cells along the fibers (Fig. 5h-k) and gait behavior (Fig. 6d) for all three conditions (Gpc4, Sdc2 and Sdc4). However, they performed time-lapse imaging, EM and assessed the number of surviving neurons only for Sdc2-enriched fibers. Gpc4 and Sdc4 are also not discussed in the Results section.

6. The number of NeuN+/GFP+ in the injured cortex implanted with Sdc2 containing gelatin fabric is very variable and seems to be skewed toward higher value by one sample. The authors may want to confirm these data with additional observations and/or analysis of Gpc4 and Sdc4 brains.

Minor Points:

1. What is the measure for the Figure 6c? Is it a percentage or cell density?

2. The authors should describe in more details experimental conditions for optogenetic stimulations (laser power, illumination time, etc)

Reviewer #3 (Remarks to the Author):

The manuscript entitled "Identification of a growth cone as the probe and driver of neuronal migration in the injured brain", Najajima et al. compared traditional axonal growth cones with the leading process of migrating neurons, which were isolated from the V-SVZ or glanglionic eminences. They showed that the terminals of migrating neurons contain similar cytoskeletal elements (actin and MTs) and receptors as cortical axonal growth cones. Moreover, these leading processes of migrating neurons respond to inhibitory cues similar to growth cones and are driven by polymerization of actin filament and MTs. Several experiments were run to show that terminals of migrating neurons respond to inhibitory environments in vitro and in vivo. Finally, the authors showed that gelatin fiber nanowoven fabric loaded with HSPGs could promote the migration of neurons and behavior recovery from injury.

While I find that the data and figures in this paper to be high quality, I feel the significance and novelty of this work is not sufficient to warrant publication in Nature Communications. It has been known for decades that the terminals of migrating neurons provide the motile force to power cell migration. While this report validates that terminals of migrating neurons are in fact growth cones, this observation is not unexpected. Figures 1-4 are largely descriptive studies demonstrating that terminals of migrating neurons look and behave like axonal growth cones. The second half of this report switches to models of regeneration, which I find to be somewhat disconnected from the first set of observations. Engineered biomaterials that promote axon regeneration and cell migration have been examined for year, and it difficult to know if their gelatin-fiber nanowoven fabric is marked improvement from past methods.

Responses to the Reviewers

Reviewer #1 (Remarks to the Author):

In this manuscript, Nakajima et al. present diverse sets of data showing the role of a growth-cone-like structure at the leading process of a migrating neuron (LP growth cone), particularly of its leading filopodium as an important structure that translates environmental molecular information into translocation of the neuronal soma. Furthermore, their research on LP growth cones has contributed to the development of a new type of biomaterial containing heparan sulfate proteoglycans (HSPGs) that facilitates neuronal migration and functional recovery after brain cortical injury. The experiments in this paper have been carefully designed, and its main conclusions are well supported by the experimental data. I think this work contributes substantially to our better understanding of cellular mechanisms underlying neuronal migration in the injured brain. I only have a few minor comments as follows:

1. Probably the most apparent difference between axonal growth and LP growth is that the latter must be accompanied by somal translocation. As explained in page 4 and shown in Figure 1b, migrating neurons exhibit alternating LP growth and pause phases with somal pausing and translocation. It seems to me that the soma moves forward when its LP is in the pause phase. Is this statistically true? If so, it would make for an interesting discussion on the presence of intracellular signaling that mediates the synchronized behavior of the LP growth cone and the soma.

We appreciate for the reviewer's insightful suggestion. We analyzed the somal speed of cultured migrating neurons expressing Venus-CAAX during leading-process (LP) elongation and pausing phases. We indeed found that the somal speed in the LP pausing phase was significantly higher than that in the LP elongation phase, indicating that individually migrating V-SVZ-derived neurons show somal pausing and translocation during LP elongation and pausing phases, respectively. The conclusion corresponds with the reviewer's insight. These phenomena are consistent with those observed for chain-forming V-SVZ-derived neurons in the rostral migratory stream (Ota et al., *Nat Commun* 2014; Matsumoto et al., *J Neurosci* 2019). The new results were included in Fig. 1c of the revised manuscript and described in the Results section as follows:

“While the axons of cultured differentiating neurons derived from embryonic cerebral cortex elongated (Fig. 1a), cultured migrating neurons derived from the neonatal V-SVZ showed alternating LP elongation and pause phases with somal pausing and translocation, respectively (Fig. 1b, c), a pattern that resembles that exhibited by chain-forming migrating neurons in the rostral migratory stream (RMS)²⁵.” (Page 4, lines 8–13)

Several previous studies have suggested intracellular mechanisms that may underlie synchronized LP behaviors and somal dynamics in migrating neurons, including ATP, cAMP, and Ca²⁺ (Guan et al., *Cell* 2007; Stoufflet et al., *Sci Adv* 2020; Bressan et al., *eLife* 2020). Given our observation that LP growth cone dynamics were synchronized with somal ones, it is possible that growth cone dynamics regulated by PTPσ-mediated machineries regulate saltatory neuronal migration by affecting ATP, cAMP, and Ca²⁺. These possibilities are now mentioned in the Discussion section as follows:

“Recent studies have implicated ATP, cAMP, and Ca²⁺ in the LP-soma dynamics in migrating neurons^{49–51}. Thus, growth cone dynamics controlled by PTPσ-mediated cytoskeletal reorganization may regulate saltatory neuronal migration by affecting these intracellular signaling molecules.” (Page 11, lines 24–27)

2. In the first paragraph in page 6, I assume that the authors intend to compare neuronal behavior on HSPG versus control stripes. However, it is unclear which descriptions are about neurons on HSPG stripes.

We apologize for providing insufficient detail in our descriptions of HSPG stripes in Fig. 2. Indeed, we intended to compare the dynamics of leading tip structures on Sdc2 stripes with those on control ones. For clarity, we added “on Sdc2 stripes” to the Results section as in the following:

“Similar to axonal growth cones³⁶, LP growth cones collapsed and failed to extend through CSPG-containing matrigel on control stripes (Fig. 2b, f, g) but each produced a singular filopodium with an elongating leading tip on Sdc2 stripes (Fig. 2c, arrows) that showed PTPσ expression (Supplementary Fig. 3c). These leading filopodia were longer than other non-tip filopodia (Fig. 2d). Following elongation of each leading filopodium, a lamellipodia-like section of plasma membrane extended in the same direction as the leading filopodium’s elongation, forming a typical-appearing growth cone on Sdc2 stripes (Fig. 2c, 5–9 min; Fig. 2e) that moved anteriorly and was succeeded by somal translocation (Fig. 2c, 5–29 min; Fig. 2f, g).” (Page 6, lines 5–14)

3. It is stated in page 10, “Our photoinactivation experiments suggested that the proper balance of F-actin and microtubule assembly dynamics is essential for leading filopodium elongation.” However, I am not sure what exactly this “balance” means. Also, I cannot find any experiments that examined the balance of cytoskeletal dynamics.

We deeply apologize for our confusing description. As the reviewer pointed out, our data did not show any “balance” of actin and microtubule assembly dynamics. Rather, our photoinactivation experiments strongly suggest the importance of “spatiotemporal regulation” of cytoskeletal dynamics in leading filopodium elongation. Therefore, we revised this sentence as follows:

“Our photoinactivation experiments suggested that proper spatiotemporal regulation of F-actin and microtubule assembly dynamics is essential for leading filopodium elongation and further suggested that lamellipodia extension and cell body movement are strongly dependent on this process.” (Page 12, lines 7–10)

Reviewer #2 (Remarks to the Author):

In this manuscript, Nakajima and co-authors assessed the cytoskeletal dynamics of growth cones of migrating cells and their responsiveness to chondroitin sulfate (CS) proteoglycans through protein tyrosine phosphatase receptor type sigma (PTP σ). The authors further studied whether heparan sulfate (HS)-containing biomaterials implanted into the injured cortex attenuate the inhibitory effects of CS and promote neuronal migration and recovery. The authors used compelling experimental approaches, and their findings are interesting. Specifically, using super-resolution and time-lapse imaging approaches, the authors showed that growth cone of migrating cells display structural and cytoskeletal dynamics which is similar to that of axonal growth cone. They also showed that growth cone responsiveness to some micro-environmental cues depends on the same molecular pathways as in the axonal growth cones and that modulation of these pathways either ontogenetically or through HS-containing gelatin fibers affects the efficiency of neuronal migration in vitro, as well as in vivo, in the injured cortex, to promote the functional recovery.

Together these results provided a detailed characterization of growth cone cytoskeletal dynamics and revealed some of CS- and HS-activated molecular pathways. This is an interesting and comprehensive study that may be strengthened by addressing following points.

1. The authors performed a comprehensive analysis (Fig. 1; Supp Fig 1) for the expression of multiples molecular factors in the growth cone of migrating neurons. They selected factors that were reported to be enriched in the growth cone of axons. However, essentially all factors, except two, were not enriched in the growth cone of migrating neurons (Supp. Fig. 1K). The authors have used machine learning-based algorithms for the extraction of growth cones of migrating cells and calculated the intensity ratio of immunofluorescent signals between the growth cone and the shaft of the leading process. Since these approaches have not been used for axonal growth cone, it would be informative to perform similar analysis for some of these molecular cues in the axons. It will allow side-by-side comparison of molecular signatures of growth cones of axons and migrating cells, and will be in line with other comparative analysis reported by authors.

We appreciate the reviewer’s constructive comments. We first established a machine learning-based algorithm for the extraction of axonal growth cones that are defined as a membrane extension at the tip of a Dcx-EGFP⁺ axon (for methodological details, please see

the Methods section of the revised manuscript). As shown in Fig. 1g, we were able to extract EGFP+ axonal growth cones with our algorithm.

We next examined whether our algorithm could extract LP growth cones in Dcx-EGFP+ migrating neurons. We found that GFP+ growth cones could be extracted automatically and unambiguously with our algorithm, strongly supporting the notion that LP growth cones are morphologically analogous to axonal ones. Furthermore, this algorithm enabled us to evaluate the enrichment of axonal growth cone molecules in LP growth cones, and it revealed that PTP σ , Liprin- α , Destrin, and Nrdg1 are enriched in both axonal and LP growth cones (revised Fig1. f–j; Supplementary Fig. 2a).

We now believe that our algorithm for machine learning-based growth cone extraction allows side-by-side comparison of molecular signatures of axonal and LP growth cones as was suggested by the reviewer. Furthermore, we agree with the reviewer’s opinion that these molecular comparative analyses are in line with our morphological and dynamic ones, strongly increasing the novelty of this study. We included these additional comprehensive analyses in Fig. 1f–j of the revised manuscript and the findings are now described in the Results section as follows:

“For this purpose, we established a machine learning-based algorithm for extracting the axonal growth cones of Dcx-EGFP+ neurons (Fig. 1f–j). This algorithm enables the automatic detection of EGFP+ axonal growth cones (Fig. 1g). Applying the algorithm to Dcx-EGFP+ migrating neurons enabled us to demarcate the growth cones of LPs (Fig. 1h), consistent with the supposition that LP growth cones are analogous to axonal ones. We calculated enrichment of axonal growth cone molecules as the ratio of each immunopositive signal within extracted LP growth cones relative to that in adjacent leading shafts (Supplementary Fig. 2a). The CSPG/HSPG-responsive receptor PTP σ , the scaffold protein Liprin- α , the actin depolymerizing factor Destrin, and N-myc downstream regulated-1 (Ndrg1) were observed to be enriched in both LP growth cones and axonal growth cones (Fig. 1i, j; Supplementary Fig. 2a, b).” (From page 4, line 31 to page 5, line 5)

2. Some data reported by the authors are based on observations from low number of cells. In some experiments, the authors used only 3-5 cells (Figs. 1-3). The number of cells is very low and should be increased, since it may not be representative of the entire population of migrating cells.

According to the reviewer’s suggestion, we performed additional timelapse imaging and increased analyzed cell numbers up to at least 10 cells (at least three independent experiments) in Figs. 1l, 2e, f, 2j–m, 3h–j, and supplementary Fig. 3d–e of the original manuscript. The conclusions of each experiment did not change after adding cells, and we think that our results are representative of the migrating neuron population at large. Furthermore, we performed new timelapse imaging (Fig. 2l–p, 3b, Supplementary Fig. 4d–f of the revised manuscript) in which the analyzed cell quantities were 10–15 cells per experimental group.

These new results were included in Fig. 1m–r, 2b–g, 2l–s, 3b, 3g–i, Supplementary Fig. 1j–q, 4d–g, 5b–d of the revised manuscript, and the analyzed cell numbers are provided in Supplementary Table S2.

3. Optogenetic experiments are interesting but requires additional controls with i) 405 nm illumination of the leading filopodium without photoswitchable inhibitors to show that blue light illumination alone does not change the dynamics of leading filopodium, and ii) demonstration that actin and microtubule dynamics were effectively affected by optogenetic stimulations in the leading process and shaft, in the presence of photoswitchable inhibitors. More information and validation are also required for opto-latrunculin, which is not yet published.

According to the reviewer’s comment, we first performed 405-nm laser-illumination of the leading filopodium in migrating neurons without photoswitchable inhibitors. We found that the leading filopodium length, growth cone extension, and speed of migrating neurons were not affected by 405-nm laser-illumination, indicating that illumination at 405 nm does not affect neuronal migration behaviors. These results were included in Fig. 3 of the revised manuscript and described in the Results section as follows:

“The control laser illuminations, 405 nm without the inhibitors or 514 nm with the inhibitors, did not inhibit leading filopodium elongation (Fig. 3b, c, e, g–i; Supplementary Fig. 5a–d; Supplementary Movie 3), suggesting that laser illumination did not cause non-specific cell damage.” (Page 7, lines 25–28)

To validate the inhibitory effects of Opto-Lat and PnOJ on actin dynamics more specifically, we introduced the F-actin indicator EGFP-UtrCH, instead of EGFP-actin and DsRed, into migrating neurons and assessed the effects of inhibitors. We found that EGFP-UtrCH⁺ intensity in the leading filopodium, but not that in the leading shaft, had decreased significantly following Opto-Lat activation. In contrast, while EGFP-UtrCH⁺ intensity was decreased during leading filopodium retraction, it was significantly sustained by PnOJ activation. Together these results suggest that photoactivation of Opto-Lat and PnOJ modulate F-actin dynamics efficiently in migrating neurons.

To examine how PST-1 inhibits microtubule dynamics, we labeled microtubule plus ends with EB3-EGFP and found that the density of EB3-EGFP⁺ dots in the leading filopodium was significantly decreased after PST-1 activation, suggesting that PST-1 can suppress microtubule polymerization efficiently in migrating neurons.

These new results are included in the revised Supplementary Fig. 4d-g and mentioned in the Results and Methods sections as follows:

“Leading filopodium tips on Sdc2 stripes were illuminated with a 405-nm laser to inhibit actin and microtubule dynamics, while the proximal LPs adjacent to the tips were

illuminated with a 514-nm laser to suppress the activity of potentially diffusing activated inhibitors (Fig. 3a; Supplementary Fig. 4d–g).” (Page 7, lines 21–24)

“On the other hand, in the presence of the actin depolymerization inhibitor PnOJ (activated), the leading filopodium was maintained, suggesting that F-actin depolymerization is involved in leading filopodium retraction (Supplementary Fig. 4d–g).” (Page 7, lines 30–33)

“Concentrations of photoswitchable inhibitors were determined based on the observation of inhibitory effects of actin and microtubule dynamics (Supplementary Fig. 4d–g) and a previous report⁶³.” (Page 25, lines 13–15)

We apologize to the reviewer for not being able to include methodological Opto-Lat details in the original manuscript. Those details are described in the following dissertation, which is currently available for review only: Nynke Anna Vepřek, The Photopharmacology of Actin and Cytoskeletal Regulators and De novo Design of SARS-CoV-2 Main Protease Inhibitors. Dissertation, Ludwig-Maximilians University, Munich (2021). We attached the corresponding chapter of the dissertation in which Optolatrunculin (Opto-Lat, **48**) is described in the appended file ‘For Review Only.pdf’. We added the source information for Opto-Lat, PnOJ, and PST-1 to the Methods section as follows:

“Opto-Lat was synthesized and characterized as described previously by Vepřek (2021, The Photopharmacology of Actin and Cytoskeletal Regulators and De novo Design of SARS-CoV-2 Main Protease Inhibitors. Dissertation, Ludwig-Maximilians University, Munich). Details on PnOJ and PST-1 are described elsewhere^{39,40}.” (Page 25, lines 9–13)

We have also assessed the inhibitory effect of Opto-Lat in our experimental system as mentioned above (Supplementary Fig. 4d, g of the revised manuscript). These new results are included in the revised manuscript.

4. The authors quantified the percentage of migrating cells along the gelatin fibers enriched with HS. They focused their analysis at the level of cell bodies of migrating cells, which is in contrast to all previous analysis showing PTP σ enrichment in the growth cone. Can the authors look at the level of growth cone? To get better cellular resolution this can be done with electroporated mice.

We thank the reviewer for this important point. Following the reviewer’s suggestion, we have added new experiments to evaluate growth cone dynamics in HS-enriched injured cortex using slice cultures. Slice culture experiments can provide data on the actual dynamics of *bona fide* growth cones within ca. 10 min, as previously demonstrated (original Fig. 5o–r; Supplementary Movie 8).

We first performed preliminary experiments in which we electroporated lateral ventricle cells in newborn mouse pups, at P0, with plasmids encoding *EmGFP* under the control of the *CAG* promoter and cryoinjured the cortex at P2. We conducted careful immunohistochemical analyses of the resultant brain sections obtained from the mice at P7. Although the electroporated neurons provide a clearly visible morphology (Figure I below), because it is possible to recognize the growth cone morphology observed in *Dcx-EGFP* brain sections (revised Figure 5q–t; Supplementary Fig. 7a; Supplementary Movie 8), we sought to use the *Dcx-EGFP* sections. The use of *Dcx-EGFP* brain slices facilitates the identification of migrating neurons while providing an abundance of target cells for LP growth cone assessment.

Next, to corroborate the *ex vivo* growth cone data we obtained from Sdc2-enriched samples, we performed new slice culture experiments on Gpc4 and Sdc4-gelatin fibers implanted into injured cortex. Briefly, slices of P7–8 *Dcx-EGFP* injured cortex inserted with Gpc4- and Sdc4-containing gelatin fibers or with gelatin fibers only were subjected to the experiments. The new results indicated that conditions with Gpc4 and Sdc4 on gelatin fibers promoted growth cone extension in CS-rich environments, while the control condition did not (revised Supplementary Fig. 7d, e; Supplementary Movie 9). These new results were consistent with our previously obtained result from Sdc2 samples. From the above results, we can conclude that HSPGs promote migration and also increase the frequency of growth cone extension *in vivo*. We described these new data in the Results and Figure legends as follows:

“Additional quantitative analysis of slice culture images of injured cortex implanted with Gpc4- or Sdc4-loaded GFs revealed significantly more frequent growth cone extension after contact with Gpc4- or Sdc4-enriched GFs than after contact with control GF (Supplementary Fig. 7d, e; Supplementary Movie 9).” (Page 10, lines 16–20)

*“Supplementary Movie 9: Time-lapse imaging of the slice cultures from P7 *Dcx-EGFP* cryoinjured cortex with implanted Gpc4 or Sdc4-containing GFs*

Migrating neurons extended their growth cones after contacting gelatin fibers enriched with Gpc4 or Sdc4; growth cones stayed collapsed after contacting control gelatin fibers.” (Description of Additional Supplementary Files)

5. The rationale for including Gpc4 and Sdc4 in some experiments, but not in others is not clear. For example, the authors reported the distribution of Dcx-GFP+ cells along the fibers (Fig. 5h-k) and gait behavior (Fig. 6d) for all three conditions (Gpc4, Sdc2 and Sdc4). However, they performed time-lapse imaging, EM and assessed the number of surviving neurons only for Sdc2-enriched fibers. Gpc4 and Sdc4 are also not discussed in the Results section.

We apologize for not adequately including the Gpc4 and Sdc4 data in the original manuscript. To address the reviewer’s comment, we have performed additional experiments that showed similar outcomes for migrating neurons across all three HSPGs (Gpc4, Sdc2 and Sdc4). We added new time-lapse imaging data from the slice culture experiments as mentioned above (point 4; revised Supplementary Fig. 7d, e; Supplementary Movie 9), and quantitative data from mature neurons in injured cortex (revised Supplementary Fig. 7f) after implantation of Gpc4-, Sdc2, or Sdc4-enriched gelatin fibers.

We performed new time-lapse slice culture experiments examining the effect of Gpc4- and Sdc4-loaded GF on growth cone extension as mentioned above (point 4). We have described these new data in the Results as follows:

“Additional quantitative analysis of slice culture images of injured cortex implanted with Gpc4- or Sdc4-loaded GFs revealed significantly more frequent growth cone extension after contact with Gpc4- or Sdc4-enriched GFs than after contact with control GF (Supplementary Fig. 7d, e; Supplementary Movie 9).” (Page 10, lines 16–20)

To obtain additional quantitative data on surviving neurons, we prepared samples with Gpc4 and Sdc4 gelatin fibers or with gelatin fibers only, as previously described for Sdc2 samples in the original manuscript. We quantified NeuN+EmGFP+ cells at P30 in injured cortex (point 6 and specific point 1; revised Supplementary Fig. 7f). The number of EmGFP-labeled (V-SVZ-derived) cells expressing NeuN was significantly greater in Gpc4- and in Sdc4-inserted samples than in the control samples (Supplementary Fig. 7f). By following the reviewer’s comment, we were able to include important data leading to the conclusion that the samples loaded with all three HSPGs (Gpc4, Sdc4 and Sdc2) contain more V-SVZ-derived NeuN+ neurons in the injured cortex. We have described these new data in the Results as follows:

“Significantly larger numbers of NeuN+EmGFP+ mature neurons were observed in brains implanted with HSPG-loaded GF than in brains with control GF (Fig. 6b, c; Supplementary Fig. 7f).” (Page 10, lines 29–31)

We have also attempted to obtain the TEM images of neurons directly attached to the Gpc4- or Sdc4-enriched GFs. After careful consultation with Prof. García-Verdugo, an expert in electron microscopy of new neurons and a co-author, we decided not to include the newly obtained images in the revised manuscript for the following reason. Given the large number of neurons migrating along the GF containing the HSPGs (Fig. 5i–k), it is likely that TEM sections we observed also contain new neurons. However, the morphology and cytoplasmic characteristics of new neurons in the injured brains differ from those in intact brains, making it difficult to confidently identify new neurons by TEM. Only TEM images that have been carefully confirmed to be new neurons based on the strict criteria were selected and included in the manuscript (Fig. 5l, m).

In the original manuscript, we showed that when new neurons attach to the Sdc2-loaded gelatin fibers, their migrating speed increases (Fig. 5e–g; Supplementary Fig. 6e; Supplementary Movie 7). We have now added new *in vitro* data indicating that new neurons increase the migrating speed when they attach to gelatin fibers loaded with HSPGs (Sdc2, Sdc4, or Gpc4) compared to those migrating on the control gelatin fibers (revised Supplementary Fig. 6f). Taken together, these results suggest that new neurons directly attach to the GF, resulting in increased migration speed *in vivo*. We have described these new data in the Results as follows:

“Time-lapse imaging showed that cultured neurons migrated faster on GF that had been loaded (by way of immersion) with Sdc2, Sdc4, or Gpc4 (the HSPGs used in the above experiments) than on control GF without the HSPG (Fig. 5e–g; Supplementary Fig. 6e, f; Supplementary Movie 7).” (Page 9, lines 16–19)

6. The number of NeuN+/GFP+ in the injured cortex implanted with Sdc2 containing gelatin fabric is very variable and seems to be skewed toward higher value by one sample. The authors may want to confirm these data with additional observations and/or analysis of Gpc4 and Sdc4 brains.

We agree that there were large differences among samples with Sdc2-containing GF, even with the same sample preparation method. The variance was due mainly to the difficulties associated with controlling injury extent and the precise location of implanted materials, factors that are critical for facilitating neuronal migration. In the revised manuscript, we added 4 Sdc2 and control samples each; the updated average values (control, 5.40 ± 0.763 ; Sdc2, 10.67 ± 2.58) were similar to the averages reported in the previous manuscript data (control, 4.14 ± 0.776 ; Sdc2, 10.31 ± 2.17). The repeated experiments provided a wider distribution of data, but no change in the conclusion. The new data (revised Fig. 6c) confirmed the previous data. Furthermore, as suggested, we performed the experiments with Gpc4 and Sdc4 and obtained results similar to those obtained with Sdc2 (revised Supplementary Fig. 7f). Given that Gpc4, Sdc2 and Sdc4 gave the same results, we are confident that HSPGs promote neuronal regeneration. We added text to the Results section as follows:

“Significantly larger numbers of NeuN+EmGFP+ mature neurons were observed in brains implanted with HSPG-loaded GF than in brains with control GF (Fig. 6b, c; Supplementary Fig. 7f).” (Page 10, lines 29–31)

1. What is the measure for the Figure 6c? Is it a percentage or cell density?

We appreciate the careful reading of our original manuscript and sincerely apologize for omitting this analytical detail. The measure in Figure 6c is the number of NeuN+EmGFP+ co-expressing cells counted from every second coronal section. We multiplied the obtained raw data by two and show the resultant quantity on the graph in the revised manuscript as the number of total cells from the injured cortex. We have amended the figure legend and added the quantitative analysis method to the Methods as follows:

“(c) Quantification of EmGFP+NeuN+ cells in injured cortex. The graph shows total number of EmGFP-labelled NeuN cells in P30 control or Sdc2-containing gelatin fabric-implanted injured cortices. A dot represents examined brain.” (Page 19, lines 21–23)

“The number of NeuN+EmGFP+ co-labelled cells in every second section containing the lesion track was counted and the cell count number was multiplied by two to obtain the estimated number of total cells per injured cortex sample.” (Page 28, lines 27–30)

2. The authors should describe in more details experimental conditions for optogenetic stimulations (laser power, illumination time, etc)

We apologize for providing insufficient descriptions of the experimental conditions of the optogenetic stimulations. We added the following clarifying information about timelapse imaging and photo-illumination to the Methods:

“The fiber outputs of the 405-nm and Argon (500-nm) lasers in our microscope system were 15.08 mW and 12.33 mW, respectively. Opto-Lat, PnOJ, and PST-1 were photoswitched by 405-nm (2.0%) and 514-nm (2.0%) laser-illumination for 6–8 s in a squared ROI (4 μ m-width) set on Sdc2 stripes. Z-axis focus was maintained with the Definite Focus 2 function and auto-adjusted by illuminating IR laser at the beginning of each imaging session to ensure precise spatial photo-illumination. The imaging and illumination settings in the photo-illumination experiments were as follows: imaging scaling X/Y, 0.050/0.050 μ m per pixel; z-interval, 1.0 μ m; Zoom 2.0; pixel dwell, 0.98 μ s; average, 1; and bleaching iteration, 20.” (Page 25, lines 1–9)

Reviewer #3 (Remarks to the Author):

The manuscript entitled “Identification of a growth cone as the probe and driver of neuronal migration in the injured brain”, Najajima et al. compared traditional axonal

growth cones with the leading process of migrating neurons, which were isolated from the V-SVZ or glanglionic eminences. They showed that the terminals of migrating neurons contain similar cytoskeletal elements (actin and MTs) and receptors as cortical axonal growth cones. Moreover, these leading processes of migrating neurons respond to inhibitory cues similar to growth cones and are driven by polymerization of actin filament and MTs. Several experiments were run to show that terminals of migrating neurons respond to inhibitory environments in vitro and in vivo. Finally, the authors showed that gelatin fiber nanowoven fabric loaded with HSPGs could promote the migration of neurons and behavior recovery from injury.

While I find that the data and figures in this paper to be high quality, I feel the significance and novelty of this work is not sufficient to warrant publication in Nature Communications. It has been known for decades that the terminals of migrating neurons provide the motile force to power cell migration. While this report validates that terminals of migrating neurons are in fact growth cones, this observation is not unexpected. Figures 1-4 are largely descriptive studies demonstrating that terminals of migrating neurons look and behave like axonal growth cones. The second half of this report switches to models of regeneration, which I find to be somewhat disconnected from the first set of observations. Engineered biomaterials that promote axon regeneration and cell migration have been examined for year, and it difficult to know if their gelatin-fiber nanowoven fabric is marked improvement from past methods.

We deeply apologize for not communicating the novelty of the study clearly in the original manuscript. At the same time, we appreciate the valuable comments and the opportunity to improve the content. As suggested by the reviewer, we have amended the manuscript by making the following four changes; 1) adding to the novelty of the study, 2) reducing the descriptive parts, 3) showing consistency between the first and the second datasets, and 4) reinforcing the biomaterial experiments.

1) Additional novelties to the study

We added new experiments (Fig. 1-3) to increase the novelty of the first half of the manuscript.

In Fig. 1, to increase the novelty of the findings regarding the commonality between axonal and LP growth cones, we performed comprehensive screening of the localization of axonal growth cone molecules in LP growth cones by introducing a machine learning-based algorithm for axonal growth cone extraction. The algorithm extracted LP growth cones successfully, strongly supporting the morphological commonality between axonal and LP growth cones. Our screening revealed that in addition to PTP σ and Liprin- α , Destrin (actin-depolymerizing protein) and Nrdg1 (N-myc downstream regulated-1) are also concentrated in both axonal and LP growth cones, further strengthening the molecular commonality between axonal and LP growth cones (revised Fig. 1g-j, Supplementary Fig. 2a).

As shown in Fig. 2, employing a proteomics-based approach, we identified Cortactin as a protein enriched in the LP growth cone (revised Fig. 2i, j). Considering the previous reports on axonal growth cones indicating that Cortactin is a substrate of Src (He et al., *Mol Biol Cell*, 2015) and is dephosphorylated at its tyrosine 421 residue (Y421) by PTP σ (Sakamoto et al., *Nat Chem Biol*, 2019), we investigated the localization, regulation, and function of Cortactin phosphorylated at Y421 in LP growth cones. We found that Y421-phosphorylated Cortactin is enriched in LP growth cones, similar to that previously reported in axonal growth cones (Sakamoto et al., *Nat Chem Biol*, 2019) and that Y421-phosphorylated Cortactin is drastically reduced by fyn-KD (revised Fig. 2k), suggesting that Fyn acts as a kinase that phosphorylates Cortactin at Y421, which has not been reported in axonal growth cone. Although there are reports on the above-mentioned findings regarding the expression of Cortactin in axonal growth cones, there are no reports on the function of pY421-Cortactin in regulating growth cone morphology. We then found that fyn- or cortactin-KD inhibited leading filopodium elongation, growth cone extension, and somal translocation of migrating neurons in the presence of CS. The defects caused by cortactin-KD were rescued by expressing KD-resistant cortactin cDNA, but not a Y421A-mutated form of Cortactin (revised Fig. 2l-s). Here, we provide direct evidence that phosphorylated Cortactin is essential for growth cone extension and neuronal migration. This data suggests that axonal growth cone extension may also be regulated by pY421-Cortactin. These findings provide evidence for a novel regulatory mechanism and its functional significance for growth cone extension and neuronal migration, increasing the novelty and relevance of this paper considerably.

As can be seen in Fig. 3 of the revised manuscript, to increase the reliability of our photo-illumination experiments, we added several strict controls (405 nm illumination without photoswitchable inhibitors, and evaluation of inhibitors' effects with F-actin and microtubule polymerization indicators) (revised Fig. 3b, g-i and Supplementary Fig. 4d-g). These new results strengthen the conclusions of the photo-illumination experiments, demonstrating leading filopodium functionality.

The new results are described in the Introduction, Results and Methods sections as follows:

“In the presence of CS, the growth cones can revert to their extended morphology when their leading filopodia interact with heparan sulfate (HS), which induces tyrosine phosphorylation of Cortactin, thus re-enabling neuronal migration.” (Page 2, lines 9-12)

“For this purpose, we established a machine learning-based algorithm for extracting the axonal growth cones of Dcx-EGFP+ neurons (Fig. 1f-j). This algorithm enables the automatic detection of EGFP+ axonal growth cones (Fig. 1g). Applying the algorithm to Dcx-EGFP+ migrating neurons enabled us to demarcate the growth cones of LPs (Fig. 1h), consistent with the supposition that LP growth cones are analogous to axonal ones. We calculated enrichment of axonal growth cone molecules as the ratio of each immunopositive signal within extracted LP growth cones relative to that in adjacent leading shafts (Supplementary Fig. 2a). The CSPG/HSPG-responsive receptor PTP σ , the scaffold protein

Liprin- α , the actin depolymerizing factor Destrin, and N-myc downstream regulated-1 (Ndr g 1) were observed to be enriched in both LP growth cones and axonal growth cones (Fig. 1i, j; Supplementary Fig. 2a, b)." (From page 4, line 31 to page 5, line 5)

"To examine the molecular mechanisms mediating growth cone extension and neuronal migration, we performed proteomic analyses of dissected cerebral cortex and RMS tissues from postnatal day 0 (P0), when differentiating and migrating neurons are enriched. A subset of proteins with similar relative proportions of expression in cortex and RMS was identified with the gene ontology terms growth cone and neuronal projection development. Thus identified candidate proteins included an actin-related factor (cortactin), microtubule-related factors (Tctex-1, myosin heavy chain-10, and LIS1) and an RNA binding protein (HuD).

Phosphorylated cortactin at tyrosine 421 residue (pY421) is a PTP σ substrate that is enriched in axonal growth cone³⁸. Cortactin localization, regulation, and function in LP growth cones are unknown. We found that pY421-cortactin is enriched in LP growth cones (Fig. 2j) and leading filopodia (Supplementary Fig. 3j), and that its expression is reduced following the addition of PP2, a Src-family inhibitor (Supplementary Fig. 3k), or fyn-KD (Fig. 2k), suggesting that Fyn influences pY421-cortactin in LP growth cones. Moreover, fyn- or cortactin-KD caused defects in leading filopodium elongation, growth cone extension, and somal translocation of migrating neurons (Fig. 2l–n, q–s), suggesting that Fyn and cortactin are involved in these processes.

To examine the role of pY421-cortactin in growth cone extension and neuronal migration, we introduced KD-resistant cortactin (cortactin) and its Y421A mutant (cortactin*Y421A) into cortactin-KD neurons. The defects induced by cortactin-KD were rescued by expressing cortactin* but not cortactin*Y421A (Fig. 2o–s; Supplementary Fig. 3l). These results suggest that pY421-cortactin, whose activity is reciprocally regulated by Fyn and PTP σ , is critical for extension of leading filopodia and growth cones, and thus regulates neuronal migration in the presence of CS (Supplementary Fig. 3m)."* (From page 6, line 19 to page 7, line 7)

"The control laser illuminations, 405 nm without the inhibitors or 514 nm with the inhibitors, did not inhibit leading filopodium elongation (Fig. 3b, c, e, g–i; Supplementary Fig. 5a–d; Supplementary Movie 3), suggesting that laser illumination did not cause non-specific cell damage." (Page 7, lines 25–28)

"Leading filopodium elongation into the 405 nm-illuminated region was hindered by inhibition of actin or microtubule polymerization with Opto-Lat and PST-1, respectively (Fig. 3c–g; Supplementary Movie 3)." (Page 7, lines 28–30)

"It is likely that LP growth cones have downstream mechanisms in common with axonal growth cones, particularly those involving Liprin- α ³⁵ and Cortactin³⁸." (Page 11, lines 22–24)

"Liquid chromatography /mass spectrometry of mouse tissues for proteomics Proteomics analysis was performed as previously reported^{31,34}. Briefly, cerebral cortex and RMS tissues from P0 pups were dissected, freshly frozen in liquid nitrogen, and stored in a

deep freezer.... The proportion of total cells that were Dcx+ in the RMS tissues was $81.6 \pm 6.9\%$ ($n = 3$ pups). Growth cone-related proteins were extracted with the following parameter settings: (1) normalized spectral abundance factor (NSAF): $0.8 < \text{cortex/RMS ratio} < 1.25$; (2) gene ontology (cellular component): growth cone; (3) gene ontology (biological process): neuronal projection development. Gene ontology analysis was performed using the PANTHER classification system (pantherdb.org).” (From page 25, line 17 to page 26, line 15)

In addition to adding the above new results in the first half of the study, we clarified our findings by updating the text as follows:

“A full mechanistic understanding of neuronal migration and regeneration will require clarifying the significance of morphological changes in the GCLSs of migrating neurons and elucidating both the extracellular and intracellular molecular mechanisms that regulate them.” (Page 3, lines 1–4)

“Using an artificial HSPG-containing scaffold, we succeeded in promoting growth cone-mediated neuronal migration, regeneration of mature neurons, and functional recovery in a mouse brain-injury model. Based on these findings, we propose neuronal migration mechanism wherein growth cones regulate migration through interaction with extracellular environments.” (Page 3, lines 31–36)

“The present findings indicate that neuronal migration is mediated by functional growth cones whose activities are regulated by extracellular and intracellular molecules through $PTP\sigma$.” (Page 11, lines 30–32)

“This study demonstrated the existence and functionality of growth cones in migrating neurons and produced evidence of molecular mechanisms underlying that migration, thus contributing to our understanding of how neuronal migration is regulated in response to the surrounding extracellular milieu⁶¹, the composition of which evolves during development and is altered in adult brains by disease or injury.” (Page 13, lines 17–21)

“Elucidation of the molecular mechanisms that mediate growth cone interactions with the local extracellular environment and thereby promote growth cone extension will enable the development of novel regeneration technologies based on the promotion of neuronal migration.” (Page 13, lines 23–27)

2) Reduction of the descriptive parts

We removed descriptive parts, re-structured the manuscript, and shortened the overall length. Specifically, the results on leading process growth cones, which show equivalent mechanisms to those reported in axons, were removed from the main figures. The imaging analyses and cytoskeletal dynamics scheme (original Figure 1 e–l; Figure 3 a, b, k) were placed in supplementary figures (revised Supplementary Figs. 1, 3 and 4).

3) Consistency in the first and the second sets

To better link the first and second parts of the study, we modified the texts as follows:

“The in vitro experiments showed that the growth cone dynamics are important for the neuronal migration. We next performed histological analysis in a mouse model of brain injury to examine whether in vivo neuronal migration employs the growth cone regulatory mechanisms that we observed in vitro.” (Page 8, lines 9, 10)

“Given the involvement of proteoglycans, PTP σ , and cortactin in regulating growth cone extension, we hypothesized that modulating the extracellular environment in injured cortex may affect neuronal migration by altering growth cone dynamics.” (Page 8, lines 34–36)

Providing more information about the LP growth cone morphology in the injured brain may better connect the context to that of *in vitro* growth cone observations. Therefore, we have added interactive 3D models of the obtained SBF-SEM images to the manuscript. The 3D models provide more information on the LP growth cone morphology than the 2D images (Fig. 4f, g). The URL links for the 3D models are described in the figure legends as follows:

“(f, g) Representative three-dimensional constructions of neurons (green) in the P9 injured cortex analyzed by SBF-SEM. Yellow represents nuclei. Extended (f) and collapsed (g) growth cones are shown. Boxed areas are enlarged in f¹ and g¹. Arrows and arrowheads indicate filopodia and lamellipodia, respectively. Interactive 3D models of neurons are shown at <https://sketchfab.com/3d-models/migratory-neuron-with-extended-growth-cone-b6c4b616f56343929cab8e3edca1c884> (f) and <https://sketchfab.com/3d-models/migratory-neuron-with-collapsed-growth-cone-70648b036df64a01a30339717b22537f> (g) (password: [saisei8532](https://sketchfab.com/3d-models/migratory-neuron-with-collapsed-growth-cone-70648b036df64a01a30339717b22537f)).” (Page 17, lines 22–29)

Furthermore, we considered that increasing our *in vivo* analyses of migrating neuron growth cones may provide more consistency. Therefore, we added new experimental results on growth cones in the second part the manuscript (revised Supplementary Fig. 7d, e; revised Supplementary Movie 9). The additional results provided more evidence for growth cone extension after contact with HPSGs-enriched fibers in injured mouse pup cortex, employing an experimental system that resembles our *in vitro* stripe assays in the first part of the study (Fig. 2). The text was changed as follows:

“Additional quantitative analysis of slice culture images of injured cortex implanted with Gpc4- or Sdc4-loaded GFs revealed significantly more frequent growth cone extension after contact with Gpc4- or Sdc4-enriched GFs than after contact with control GF (Supplementary Fig. 7d, e; Supplementary Movie 9). These results indicated that HSPG-augmented GF is an excellent scaffold for promoting LP growth cone extension and neuronal migration.” (Page 10, lines 16–21)

4) Reinforcing biomaterial experiments

As suggested by both the reviewer and the editor, we tried our best to demonstrate the novelty of using gelatin-fiber nonwoven fabric (GF) in the promotion of neuronal migration.

We added text explaining the advantages of GF. We apologize for not fully explaining the features of the GF in the previous manuscript. We have modified the text as follows:

“To examine whether HSPG-containing substrate promotes neuronal migration into injured cortex, we employed gelatin-fiber nonwoven fabric (GF), a unique biomaterial that provides a structural scaffold for cells and has a high affinity for extracellular matrix molecules, including sulfated glycosaminoglycan^{41,42} (Fig. 5a).” (From Page 8, lines 36 to page 9, line 4)

“Notably, embryonic radial glial cells express HSPGs⁵⁵ and migrating neurons in both embryonic and postnatal injured brains use radial glial cells as scaffolds for migration^{20,56}. To mimic the broad distribution of radial glial fibers in developing cortex, it would be preferable to implant a biomaterial that covers a large area rather than to inject biomaterial such that it affects a small area around the injection site, though the latter approach has the advantage of minimizing the invasiveness of the procedure^{58,59}.” (Page 13, lines 1–7)

Additionally, we performed new comparative experiments with GF and two other biomaterials: gelatin sponge and polypropylene-fiber nonwoven fabric (PF). Gelatin sponge was made from the same gelatin as GF and formed in a 3D structure different to that of the GF. PF was fabricated from polypropylene fibers in a 3D structure like that of GF⁹. For more details, please refer to the Methods (below) and our SEM images (revised Fig. 5n). Our newly obtained data showed that Sdc2-enriched GF-implanted brain significantly increased the density of Dcx-EGFP+ neurons in implanted cortex compared to samples implanted with Sdc2-enriched gelatin sponge or PF (revised Fig. 5o). The GF with extracellular matrix Sdc2 approximated a biological environment well enough to facilitate neuronal migration. We describe these new data in the Methods, Results and Discussion as follows:

“Preparation of biomaterials

GFs were prepared as previously reported with a manufacture modification on Genocel-L⁶⁷. Briefly, gelatin powder (isoelectric point 5.0, viscosity 4.6 mPas and gelatin bloom strength 250 g; Nitta Gelatin, Inc., Japan) was immersed in distilled water and dissolved completely to obtain 37.5 w/v% gelatin solution. The solution was heated to 60 °C and ejected through a 250- μ m wide nozzle. The gelatin product was air-dried with heat until it had solidified and then subjected to dehydrothermal crosslinking at 140 °C for 12 h or at 160 °C for 24 h. Crosslinked GFs were die-cut into 5 mm \times 5 mm sheets. Gelatin sponge was prepared from 5 w/v% gelatin solution. Drops of heated gelatin solution were allowed to gelatinize at RT and then subjected to lyophilization for 60 h and brief heat-crosslinking under pressure (1×10^5 MPa) at 160 °C for 24 h. PFs were prepared as previously reported⁴¹.” (Page 26, lines 17–28)

“Scanning electron microscopy

The intrastructures of dried biomaterials were observed using a scanning electron microscope (FlexSEM 1000, Hitachi High Technologies Corp., Japan). The samples were placed on the microscope stage, fixed with conductive tape, and then coated with sputtered gold using a sputter coater (MSP-1S, Vacuum Device Inc., Japan). The secondary electron images were

obtained under the low vacuum mode (30 Pa) at 5 kV acceleration voltage.” (Page 27, lines 10–16)

“We compared neuronal migration in injured brain implanted with Sdc2-loaded GF, Sdc2-enriched polypropylene-fiber nonwoven fabric (PF)⁴¹, or gelatin sponge. Dried gelatin sponge has a honeycomb-like structure consisting of gelatin film walls with less interconnected pores, while the dried PF has a lattice-like structure of fibers similar to that of GF (scanning electron microscopy images, Fig. 5n; Supplementary Fig. 6i). Dcx-EGFP+ cell density was greater in cortices implanted with Sdc2-enriched GF than in those implanted with Sdc2-enriched PF or gelatin sponge (Fig. 5o), suggesting that GF structure and material are more conducive to neuronal migration.” (From page 9, line 31 to page 10, line 2)

“Regarding biomaterials, compared to gelatin-sponges with randomly oriented²⁰ and less interconnected pores (Fig. 5), gelatin-fabrics can better promote radial neuronal migration toward upper cortical layers because they consist of aligned fibers.” (From page 12, line 35 to page 13, line 1)

“To mimic the broad distribution of radial glial fibers in developing cortex, it would be preferable to implant a biomaterial that covers a large area rather than to inject biomaterial such that it affects a small area around the injection site, though the latter approach has the advantage of minimizing the invasiveness of the procedure^{57,58}.” (Page 13, lines 3–7)

REVIEWER COMMENTS

Reviewer #1 (Remarks to the Author):

The manuscript has been substantially revised, and I do not have any further comments.

Reviewer #2 (Remarks to the Author):

In the revised version of the manuscript the authors have performed several additional experiments and analysis to address my and other reviewers' concerns. These new data strengthen the manuscript and provide insights on some of molecular mechanisms operating at level of growth cone of migrating cells.

Overall, this is an interesting study and the authors have adequately addressed all my concerns and questions.

Reviewer #3 (Remarks to the Author):

In the revised paper by Nakajima et al., the authors have attempted to address my primary concern about the impact this study provides to the field. While I appreciate the authors efforts and the tremendous amount of work involved in this study, I still feel the impact does not rise to the level of papers published in Nature Communications. I believe this work should be published in a more specialized journal. Below I address a few specific reason for my opinion about changes made and the overall study impact.

One change the authors made was to "extract" process terminals and axonal growth cones using a machine-learning based algorithm. First, I do not like the term extract, as it implies they have performed a biochemical purification, while in reality they used a computer to create an image mask to measure fluorescence specifically within terminals. This is quite a common technique, but as described it sounds very technically advanced. This method ultimately resulted in a huge list of proteins labelled by ICC and found to be concentrated in axonal growth cones when quantified. Several molecules within that list were also concentrated in the LP of migrating neurons, but it is not clear if these are the only ones within this list. I am not sure what to conclude from this finding, but what is more concerning is that the images of labeled neurons are of low quality and really do not indicate any specific localization. This raises the question whether any of these antibodies were validated. As the authors have access to Airy disc super-resolution, it is not clear why these did not use that here. Even MT and actin images are lower quality than expected.

In Figures 1-2 the authors image neuronal terminals at inhibitory boundaries with CSPG, as described for neuronal growth cones previously (Snow et al., 1996). Here the authors suggest that inhibition is always preceded by a leading filopodium. Is this true in all cases? This would difficult to explain and interpret. Also, in some instances the authors used super-resolution live cell imaging (Airy disc scanning). In this case this approach is somewhat gratuitous (unlike previous ICC labeling), as there is no value to resolve below the diffraction limit in this experiment.

The next big change made by the authors was to introduce cortactin, a well-known F-actin binding and regulatory protein expressed in growth cones. In an attempt to be more mechanistic, they examined tyrosine phosphorylation of Y421 by fyn, a src family kinase (SFK). They claim this is first time cortactin tyrosine phosphorylation has been examined in growth cones, but several studies have examined SFK phosphorylation for over 20 years, but possibly not specifically Fyn. However, a larger concern I have is related to my previous comment about antibody specificity. They show pY421 cortactin ICC in Fig2J. However, oddly there appears to be little total cortactin overlapping with pY421 cortactin, which is difficult to explain. The pY421 cortactin looks very non-specific to me. Total cortactin is also saturated, making localization ambiguous.

Finally, the authors attempt to link the first part of their paper, which compares axonal growth cones to SVZ migrating neuron LPs, with the second half looking at CSPGs and neuronal migration into injured brain and along nanofibers. Unfortunately, I still find quite a disconnect between these two stories and that Fig. 4-6 could be a stand-alone paper.

REVIEWER COMMENTS

Reviewer #1 (Remarks to the Author):

The manuscript has been substantially revised, and I do not have any further comments.

We would like to thank the reviewer for the careful reading of the manuscript and the valuable feedback.

Reviewer #2 (Remarks to the Author):

In the revised version of the manuscript the authors have performed several additional experiments and analysis to address my and other reviewers' concerns. These new data strengthen the manuscript and provide insights on some of molecular mechanisms operating at level of growth cone of migrating cells.

Overall, this is an interesting study and the authors have adequately addressed all my concerns and questions.

We would like to thank the reviewer for the careful reading of the manuscript and the valuable feedback.

Reviewer #3 (Remarks to the Author):

In the revised paper by Nakajima et al., the authors have attempted to address my primary concern about the impact this study provides to the field. While I appreciate the authors' efforts and the tremendous amount of work involved in this study, I still feel the impact does not rise to the level of papers published in Nature Communications. I believe this work should be published in a more specialized journal. Below I address a few specific reason for my opinion about changes made and the overall study impact.

One change the authors made was to "extract" process terminals and axonal growth cones using a machine-learning based algorithm. First, I do not like the term extract, as it implies they have performed a biochemical purification, while in reality they used a computer to create an image mask to measure fluorescence specifically within terminals. This is quite a common technique, but as described it sounds very technically advanced. This method ultimately resulted in a huge list of proteins labelled by ICC and found to be concentrated in axonal growth cones when quantified. Several molecules within that list were also concentrated in the LP of migrating neurons, but it is not clear if these are the only ones within this list. I am not sure what to conclude from this finding, but what is more concerning is that the images of labeled neurons are of low quality and really do not indicate any specific localization. This raises the question whether any of these antibodies were validated. As the authors have access to Airy disc super-resolution, it is not clear why these did not use that here. Even MT and actin images are lower quality than expected.

In Figures 1-2 the authors image neuronal terminals at inhibitory boundaries with CSPG, as described for neuronal growth cones previously (Snow et al., 1996). Here the authors

suggest that inhibition is always preceded by a leading filopodium. Is this true in all cases? This would be difficult to explain and interpret.

Also, in some instances the authors used super-resolution live cell imaging (Airy disc scanning). In this case this approach is somewhat gratuitous (unlike previous ICC labeling), as there is no value to resolve below the diffraction limit in this experiment.

The next big change made by the authors was to introduce cortactin, a well-known F-actin binding and regulatory protein expressed in growth cones. In an attempt to be more mechanistic, they examined tyrosine phosphorylation of Y421 by fyn, a src family kinase (SFK). They claim this is first time cortactin tyrosine phosphorylation has been examined in growth cones, but several studies have examined SFK phosphorylation for over 20 years, but possibly not specifically Fyn. However, a larger concern I have is related to my previous comment about antibody specificity. They show pY421 cortactin ICC in Fig2J. However, oddly there appears to be little total cortactin overlapping with pY421 cortactin, which is difficult to explain. The pY421 cortactin looks very non-specific to me. Total cortactin is also saturated, making localization ambiguous.

Finally, the authors attempt to link the first part of their paper, which compares axonal growth cones to SVZ migrating neuron LPs, with the second half looking at CSPGs and neuronal migration into injured brain and along nanofibers. Unfortunately, I still find quite a disconnect between these two stories and that Fig. 4-6 could be a stand-alone paper.

Point-by-point response

1. One change the authors made was to “extract” process terminals and axonal growth cones using a machine-learning based algorithm. First, I do not like the term extract, as it implies they have performed a biochemical purification, while in reality they used a computer to create an image mask to measure fluorescence specifically within terminals. This is quite a common technique, but as described it sounds very technically advanced.

We thank the reviewer for the important comment. Following the suggestion, we have replaced the term ‘extract’ with alternative appropriate words and amended the concerned part as follows:

“Image processing and quantification using Weka Segmentation

Growth cone areas of Venus-CAAX-expressing extending axons and migrating neurons in time-lapse imaging were obtained from super-resolution imaging data with the Trainable Weka Segmentation plugin in FIJI (Supplementary Fig. 1a–f). Sixty maximum-intensity projection images from six cells (10 images/cell) were used as a training dataset to prepare an algorithm for classifying growth cone areas and other areas. For protein quantification within growth cones of migrating neurons during screening, images of ten randomly-selected cells with migratory bipolar morphology were acquired. We used the Trainable Weka Segmentation plugin in FIJI to establish an algorithm for axonal growth cone segmentation in fixed samples. Forty images of Dcx-EGFP+ axons were used as a training dataset to establish the algorithm. The Gaussian blur, Hessian, Membrane projections, Sobel filter, Difference of Gaussians, Structure, and Neighbor segmentation settings in the plugin were selected for machine-learning; and then area quantification was performed with the Analyze Particles tool in FIJI. In the screening of growth cone molecules, intensity ratio was calculated as follows: mean signal intensity in identified growth cone / mean signal intensity in leading shaft.” (Supplementary Information, page 11)

The usage of automatic cell profilers such as Weka Trainable Segmentation is indeed a well-established technique in automated segmentation of cellular regions of interest (e.g., growth cones) (Callahan, et al., *eLife*, 2019, <https://doi.org/10.7554/eLife.47837>). We amended the description in Results as follows to de-emphasize the role of Weka functions:

“For this purpose, we used a machine learning-based algorithm to segment axons into growth cones and remaining areas in Dcx-EGFP-positive neurons (Fig. 1f). Applying this algorithm to Dcx-EGFP+ migrating neurons enabled us to demarcate the growth cones of LPs (Fig. 1g), consistent with the supposition that LP growth cones are analogous to axonal ones. We calculated concentration of axonal growth cone molecules as the ratio of each immunopositive signal within identified LP growth cones relative to that in adjacent leading shafts.” (Page 4, line 36 to page 5, line 6)

2. This method ultimately resulted in a huge list of proteins labelled by ICC and found to be concentrated in axonal growth cones when quantified. Several molecules within that list were also concentrated in the LP of migrating neurons, but it is not clear if these are the only ones within this list. I am not sure what to conclude from this finding

We appreciate the opportunity to clarify our work. The purpose of performing the ICC screening assay was to find proteins concentrated in LP growth cones by selecting those whose signal intensity in growth cones was significantly higher than that in leading shafts. This screening process revealed Destrin, Liprin- α , Ndr $g1$, and PTP σ as significantly growth-cone enriched proteins (previous Fig. 1f-j and Supplementary Fig. 2a). In this revision, the specificity of the antibodies used to label these proteins was confirmed, and images showing their specific localization were presented (see point 3 below for detail, revised Fig. 1f, g and Supplementary Fig. 2a-d). Please note that Syntaxin-7 was not statistically significant in the assay (previous Supplementary Fig. 2a and revised Supplementary Fig. 2e) and is shown as an example of a molecule that was not found to be concentrated in the LP growth cone using a specific antibody (see point 3 below for detail, revised Fig. 1f, g and Supplementary Fig. 2d). As the reviewer mentioned, we screened a large number of proteins. We cannot conclude that the other proteins that did not reach statistical significance are not concentrated in the LP growth cone without validation of the antibodies. We also do not exclude the possibility that non-concentrated proteins may function in LP growth cones. To clarify the purpose of the screening assay and interpretation of the results, the text in the Discussion was revised follows:

“We have focused on Cortactin, Destrin, Liprin- α , and PTP σ as concentrated molecules in the LP growth cone. Notwithstanding, there could be other yet-to-be examined growth-cone concentrated molecules or even non-concentrated proteins that are important mediators of LP growth cone functions.” (Page 11, line 36 to page 12, line 3)

To avoid misunderstandings of the interpretation of our molecular screening assay results, molecules that were not significantly concentrated in the LP growth cone, except for Syntaxin-7 as a control, were excluded from the graph. In the present revised graph, data for Destrin, Liprin- α , and PTP σ , molecules that are significantly concentrated in the LP growth cone, are shown; GFP and Syntaxin-7 are also shown as controls (Supplementary Fig. 2e). Supplementary Table 1, which had listed many antibodies, was omitted, and only those antibodies that have been reported to show a single immunoblot band and are recommended by the manufacturer for use in immunolabeling are listed in the Methods section of the Supplementary Information sheet. Information about antibodies we used in previous studies (Nozumi et al., *PNAS*, 2009; Kawasaki et al., *iScience*, 2018; Ishikawa et al., *Mol Brain*, 2019; Okada et al., *Mol Brain*, 2021; Honda et al., *Cell Rep*, 2023) were included in the Supplementary Information sheet. The word ‘enrich’ was replaced with ‘concentrated’ and the text in Results was revised as follows:

“In brief, we found that the CSPG/HSPG receptor PTP σ is concentrated in migrating-neuron growth cones and regulates growth cone motility and neuronal migration.” (Page 3, lines 29-31)

“We conducted immunocytochemistry analyses to examine whether molecules that we showed to be concentrated in axonal growth cones³¹⁻³⁴ are also concentrated in the LP growth cones (Fig. 1f, g).” (Page 4, lines 34-36)

“We calculated concentration of axonal growth cone molecules as the ratio of each immunopositive signal within identified LP growth cones relative to that in adjacent leading shafts. The CSPG/HSPG-responsive receptor PTP σ , the scaffold protein Liprin- α , and the actin depolymerizing factor Destrin were observed to be significantly concentrated in LP growth cones (Fig. 1f, g; Supplementary Fig. 2a-e and 9a-d). PTP σ colocalized with F-actin in the peripheral domain of LP growth cones, including in filopodia and lamellipodia (Supplementary Fig. 2f). PTP σ was also found to be concentrated in the LP growth cones of migrating interneurons derived from embryonic ganglionic eminence (Supplementary Fig. 2h). These results indicate that concentrated expression of PTP σ is common to both axonal and LP growth cones. Liprin- α , a direct binding partner of PTP σ involved in PTP σ oligomerization³⁵, was found to be concentrated in LP growth cones and colocalized with F-actin (Fig. 1h). Liprin- α -knockdown (KD) reduced PTP σ signal levels in LP growth cones (Fig. 1i), suggesting that PTP σ concentration in LP growth cones is dependent on Liprin- α in migrating neurons.” (Page 5, lines 4-18)

3. I am not sure what to conclude from this finding, but what is more concerning is that the images of labeled neurons are of low quality and really do not indicate any specific localization. This raises the question whether any of these antibodies were validated. As the authors have access to Airy disc super-resolution, it is not clear why these did not use that here. Even MT and actin images are lower quality than expected.

We apologize for the low quality images. We thus prepared new immunolabeled samples to obtain higher quality images that show specific localization of molecules. Embedding cells in Matrigel reduces antibody permeability and blurs membrane morphology during image acquisition. To visualize the filopodia and lamellipodia extending from the growth cone clearly under these conditions, we worked to optimize fixation conditions. Specifically, to maintain the morphology of cell protrusions, warm 4% paraformaldehyde solution was added immediately (<5 s) after aspirating and removing the culture medium and allowed fixation to occur at 37 °C for 15 min. Consequently, we obtained super-resolution images from cells adhered to glass coverslips. To show the specific localization of Destrin, Liprin- α , PTP σ , and Syntaxin-7 clearly in the revised figures, the previous images were replaced with new improved images and those are shown together with whole migratory-cell images and pseudocolor intensity images (revised Fig. 1f, g).

To address the reviewer’s concern about antibody validation, we performed knockdown of Destrin, Liprin- α , PTP σ , and Syntaxin-7 (representative protein localized both in growth cones and leading shafts) in cultured new neurons, which we subjected to immunolabeling with antibodies targeting those respective proteins (revised Supplementary Fig. 2a-d). Molecular knockdown decreased the intensity of the antibody-stained signals in cultured new neurons significantly (revised Supplementary Fig. 2a-d). These results indicate that signals observed in LP growth cones (Destrin, Liprin- α , PTP σ) and perisomatic regions (Syntaxin-7) reflect localization of those proteins (revised Fig. 1f, g). The quantitative analyses on fluorescent signal intensities in growth cones compared to those in leading shafts are shown in Supplementary Fig. 2e. Regarding the Ndr $g1$ antibody, we could not perform further validation due to the discontinuation of the anti-Ndr $g1$ antibody product. Therefore, the Ndr $g1$ results were removed from the figures (previous Supplementary Fig. 2a, b) and from the main text as follows:

“The CSPG/HSPG-responsive receptor PTP σ , the scaffold protein Liprin- α , and the actin depolymerizing factor Destrin, and N-myc downstream regulated 1 (Ndr1) were observed to be significantly concentrated in LP and axonal growth cones (Fig. 1f, g; Supplementary Fig. 2a-e and 9a-d).” (Page 5, lines 6-9)

To improve image quality for this revision, the Airyscan super-resolution mode was used to acquire fine images of samples prepared with improved preparation methods (see above). We also replaced the previous F-actin and tubulin images mentioned by the reviewer with new and improved images (revised Fig. 1d, e, h). Although the resolution is limited by the challenging conditions, the structure of F-actin and tubulin is preserved, consistent with our live imaging observations (Supplementary Fig. 1h-q; Supplementary Movie 1). Additionally, we were able to improve the images showing colocalization of Liprin- α , PTP σ , and F-actin in the LP growth cones (revised Fig. 1h).

“Similar to axonal growth cones^{5,26,27} the GCLSs on the LPs of migrating neurons contain filopodia and lamellipodia, both of which were confirmed to be enriched with F-actin and tyrosinated tubulin, the latter being a marker for dynamic microtubules (Fig. 1d, e; Supplementary Fig. 1g).” (Page 4, lines 24-27)

4. In Figures 1-2 the authors image neuronal terminals at inhibitory boundaries with CSPG, as described for neuronal growth cones previously (Snow et al., 1996). Here the authors suggest that inhibition is always preceded by a leading filopodium. Is this true in all cases? This would difficult to explain and interpret.

We regret that our definition of leading filopodium and our description of experiments on them were unclear. As a point of clarification, we define a leading filopodium as a filopodium that leads to the future extension of a growth cone. Thus, under inhibitory conditions that do not allow growth cone extension (refer to Fig. 2b), no filopodium meets the definition of a leading filopodium. In other words, inhibition of growth cone extension is **not** preceded by the formation of a leading filopodium.

In the experiment that the reviewer commented on (previous Fig. 1n), we observed filopodia on LP growth cones when neurons migrating in Matrigel reached and entered the inhibitory boundary of the CSPG-containing Matrigel. After the filopodia of migrating neurons entered the inhibitory boundary, the filopodia shortened and, subsequently, the growth cone collapsed, as described for axons by Snow *et al.* (1996). The frequency of growth cone collapse following shrinkage of filopodia entering inhibitory boundaries (from Matrigel to CSPG-containing Matrigel; please see experimental scheme shown in revised Fig. 1j) as follows:

“To examine CSPG-PTP σ signaling effects on LP growth cone extension and subsequent neuronal migration, we analyzed how migrating neurons respond to the CSPG boundary in culture (Fig. 1j-p; Supplementary Fig. 2i). Similar to axonal growth cones (Snow et al.)^{15,37,38}, LP growth cones in contact with CSPG-containing Matrigel caused filopodia retraction and subsequent growth cone collapse [93.8 \pm 6.3% (n = 28 events from 12 cells, four independent experiments)], followed by somal deceleration (Fig. 1k, l, o, p).” (Page 5, lines 19-24)

We apologize for not providing sufficient schemes to explain the experiments shown in the previous Fig. 1m-r and Fig. 2a-h. The experimental methods shown in these previous two figures were used to analyze CSPG-induced inhibition and HSPG-induced promotion of growth cone extension in migrating neurons, respectively. To differentiate between the two experimental setups, a schematic illustrating the experiment shown in the previous Fig. 1m-r has been moved from the supplementary figure

to this main figure with some modifications (revised Fig. 1j). The other experimental scheme was also updated to improve accuracy (revised Fig. 2a and 3a).

Furthermore, because the term leading filopodium was defined based on the results of the experiment shown in Fig. 3, the term leading filopodium should not have been used in the previous Fig. 2a-h. The filopodium indicated by the white arrow in Fig. 2b should not have been identified as a leading filopodium. Also, the image of a long tip filopodium shown in the previous figure was inappropriate (previous Fig. 2b, 2 min). Such an elongated tip filopodium is only rarely observed under the CSPG-only conditions used in our study and is not representative (please refer to the revised Fig. 2d data for details). To avoid confusion, we replaced the previous Fig. 2b image with another (revised Fig. 2b and Supplementary Movie 2).

We further apologize that the experiment comparing the lengths of leading filopodium and non-tip filopodium (previous Fig. 2d) was ambiguous and the associated conclusions unclear. The length of the tip filopodium on the Sdc2 stripe and the control stripe should have been compared to demonstrate the effect of Sdc2 on tip filopodium elongation. We reanalyzed the data and corrected the figure legends to clarify the descriptions regarding the subject of the quantitative analysis (revised Fig. 2d, e). Lastly, we added a description of the definition of the leading filopodium and revised the text in Results as follows:

“These ~~leading tip~~ filopodia on Sdc2 stripes were longer than ~~other non-tip~~ filopodia on the control stripes (Fig. 2d). Following elongation of each ~~leading tip~~ filopodium, a lamellipodia-like section of plasma membrane extended in the same direction as the ~~tip leading~~ filopodium’s elongation, forming a typical-appearing growth cone on Sdc2 stripes (Fig. 2c, 5–9 min; Fig. 2e) that moved anteriorly and was succeeded by somal translocation (Fig. 2c, 5–29 min; Fig. 2f, g). The maximum extension area of the LP growth cones correlated directly with cell migration speeds (Fig. 2h). Taken together, these results suggest that HSPGs induce ~~tip leading~~ filopodium formation in migrating neurons in the presence of CSPGs and promote neuronal migration by extending growth cones.” (Page 6, lines 14-23)

“We ~~found~~ showed that pY421-Cortactin is concentrated in LP growth cones (Fig. 2j) and ~~leading tip~~ filopodia (Supplementary Fig. 3g), and that its signal is reduced following the addition of PP2, a Src-family inhibitor (Supplementary Fig. 3h), or Fyn-KD (Fig. 2k), suggesting that Fyn influences pY421-Cortactin in LP growth cones. Moreover, Fyn- or Cortactin-KD caused defects in ~~leading tip~~ filopodium elongation, growth cone extension, and somal translocation of migrating neurons (Fig. 2l–n, q–s; Supplementary Fig. 3i and 9e), suggesting that Fyn and Cortactin are involved in these processes.” (Page 6, line 34 to page 7, line 5)

“Because the elongation of a tip filopodium in contact with Sdc2 leads to the extension of an associated growth cone under inhibitory conditions, hereafter we will refer to this filopodium as the leading filopodium (Supplementary Fig. 3k). The significance of leading filopodia was examined in the following experiments.” (Page 7, lines 12-16)

“(b, c) Time-lapse images of Venus-CAAX (green)- and DsRed (red)-expressing migrating neurons on control or Sdc2 stripes (magenta) cultured in the CS-containing Matrigel. Boxed areas are enlarged at the bottom. Arrows and arrowheads indicate tip ~~leading~~ filopodium and growth cone, respectively.

(d) Length of tip ~~leading~~ filopodium on control and Sdc2 stripes.

(e) Degree of formed tip ~~leading~~ filopodium (TF) and growth cone (GC) on Sdc2 stripes.” (Page 16, lines 18-23)

“(l-s) Functional analyses of Fyn and Cortactin in neuronal migration. Time-lapse images of EmGFP (green)-expressing migrating neurons on Sdc2 stripes (magenta); the neurons were cultured in CS-

containing Matrigel. KD-resistant WT- (o) or Y421A- (p) Cortactin (Cortactin*) expressing cells are labelled with mCherry (red). Tip ~~Leading~~ filopodium length (q), growth cone area (r), and migration speed (s) are shown. Arrows indicate tip ~~leading~~ filopodium on Sdc2 stripes.” (Page 16, lines 31-36)

In summary, the response of filopodia of migrating neurons to CSPG was consistent with that described for those of axons by Snow et al. On the other hand, growth cone extension is always preceded by the elongation of a leading filopodium under inhibitory conditions. We thank the reviewer for the comments that prompted us to clarify these points.

5. In some instances the authors used super-resolution live cell imaging (Airy disc scanning). In this case this approach is somewhat gratuitous (unlike previous ICC labeling), as there is no value to resolve below the diffraction limit in this experiment.

As the reviewer pointed out, super-resolution is not necessary for live imaging of growth cones. However, the filopodium itself is a fine structure, and its distal portion in migrating neurons is typically less than 200 nm (Supplementary Fig. 4a, 4b, and 5b), which is below the diffraction limit. A long filopodium on the Sdc2 stripe could be acquired clearly with Airyscan super-resolution imaging but not with confocal imaging. Furthermore, two parallel filopodia extending from the leading process were observed occasionally (Supplementary Fig. 5b, 1 min), and these structures could be better captured with super-resolution imaging than with confocal imaging. Hence, using super-resolution microscopy in live-cell imaging of filopodia has advantages in capturing the dynamics of fine cellular structures, especially for leading filopodium analysis.

Nevertheless, thanks to the reviewer’s comment, we recognized that there were several phrases suggesting improperly that only super-resolution imaging can be used to observe cellular structures that can in fact be observed with confocal imaging. We apologize for this verbiage and have changed the wording in the Results section as follows:

~~“Time-lapse super-resolution imaging of EGFP-actin²⁸ and EB3-EGFP²⁹ (microtubule plus-end marker) revealed similar dynamics of F-actin and microtubules in both the GCLSs of elongating LPs and the growth cones of elongating axons³⁰ (Supplementary Fig. 1h–q; Supplementary Movie 1).”~~ (Page 4, lines 27-30)

~~“Super-resolution Imaging showed that PTP σ colocalized with F-actin in the peripheral domain of LP growth cones, including in filopodia and lamellipodia (Fig. 1h; Supplementary Fig. 2f).”~~ (Page 5, lines 9-11)

~~“Super-resolution Live imaging with fluorescently labeled actin and polymerizing microtubules (EGFP-actin and EB3-EGFP, respectively) confirmed that the elongating leading filopodia contain actin and microtubules (Supplementary Fig. 4a, b) similar to other filopodia of LP growth cones (Supplementary Fig. 1h–q).”~~ (Page 7, lines 20-23)

6. They claim this is first time cortactin tyrosine phosphorylation has been examined in growth cones, but several studies have examined SFK phosphorylation for over 20 years, but possibly not specifically Fyn.

We appreciate the reviewer for careful reading and agree with the opinion. Indeed, Cortactin is a target of Src family kinases including Src and Fyn (Wu et al., *Mol Cell Biol*, 1991, Huang et al., *J Biol Chem*, 2003); and Cortactin is a substrate of Src2 (He et al., *Mol Biol Cell*, 2015) in Aplysia axonal growth cones. What we have analyzed with migrating neurons was the involvement of Fyn in Cortactin tyrosine

phosphorylation in LP growth cones, which was not known even in axonal growth cones. We modified the description in Results as follows:

*“Cortactin phosphorylated at tyrosine 421 residue (pY421) is a PTP σ substrate in axonal growth cone³⁹. Cortactin is a substrate of Src family kinases inclusive of Src (Wu et al., *Mol Cell Biol*, 1991), and Fyn (Huang et al., *J Biol Chem*, 2003). It has been reported that Src2 targets Cortactin in a axonal growth cones (He et al., *Mol Biol Cell*, 2015). We showed ~~found~~ that pY421-Cortactin is concentrated in LP growth cones (Fig. 2j) and ~~tip leading~~ filopodia (Supplementary Fig. 3g), and that its ~~signal~~ expression is reduced following the addition of PP2, a Src-family inhibitor (Supplementary Fig. 3h), or Fyn-KD (Fig. 2k), suggesting that Fyn influences pY421-Cortactin in LP growth cones. Moreover, Fyn- or Cortactin-KD caused defects in ~~tip leading~~ filopodium elongation, growth cone extension, and somal translocation of migrating neurons (Fig. 2l–n, q–s; Supplementary Fig. 3i and 9e), suggesting that Fyn and Cortactin are involved in these processes.”* (Page 6, line 32 to page 7, line 5)

7. A larger concern I have is related to my previous comment about antibody specificity. They show pY421 cortactin ICC in Fig2J. However, oddly there appears to be little total cortactin overlapping with pY421 cortactin, which is difficult to explain. The pY421-Cortactin looks very non-specific to me. Total cortactin is also saturated, making localization ambiguous.

Following the reviewer’s suggestion, we validated the anti-Cortactin antibody using a Cortactin knockdown construct and performed new experiments to obtain better images of Cortactin and pY421-phosphorylated Cortactin. The results of our anti-Cortactin antibody validation experiment are shown in a new figure (revised Supplementary Fig. 3j) and the new immunolabeling images are shown in the revised figure (revised Fig. 2j). To reduce the overall Cortactin signal level and to demonstrate the specific intracellular localization of Cortactin labeling, we reduced the concentration of secondary antibodies and increased the number of washes. The Cortactin signals were concentrated in the growth cone, a portion of which overlapped with the pY421-Cortactin signals. In addition, Cortactin and pY421-Cortactin signal intensities were significantly decreased in migrating neurons transfected with the Cortactin-KD plasmids (revised Supplementary Fig. 3j). Based on these data, we conclude that the intracellular signal in the new images obtained from the anti-pY421-Cortactin antibody indicates tyrosine phosphorylation of Cortactin rather than non-specific staining.

In addition, we revised the text to change the term ‘enriched’ to ‘concentrated’ as follows:

“We ~~showed~~ that pY421-Cortactin is ~~enriched~~ concentrated in LP growth cones (Fig. 2j) and ~~tip leading~~ filopodia (Supplementary Fig. 3g), and that its expression is reduced following the addition of PP2, a Src-family inhibitor (Supplementary Fig. 3h), or Fyn-KD (Fig. 2k), suggesting that Fyn influences pY421-Cortactin in LP growth cones.” (Page 6, line 34 to page 7, line 2)

8. Finally, the authors attempt to link the first part of their paper, which compares axonal growth cones to SVZ migrating neuron LPs, with the second half looking at CSPGs and neuronal migration into injured brain and along nanofibers. Unfortunately, I still find quite a disconnect between these two stories and that Fig. 4-6 could be a stand-alone paper.

As the reviewer pointed out, it was important to improve the connection between the first and second parts of our manuscript, and we appreciate the reviewer's continued suggestions in this regard. In the previous revision, we attempted to connect the contents by inserting several sentences between paragraphs. Actually, the first *in vitro* and second *in vivo* studies were never separate, but rather had been

designed as a series of studies conducted for a common purpose. It was important to revise the manuscript to make this clear.

The common aim of all the experiments in this study is to explore novel mechanisms that promote neuronal migration in CS-rich damaged brain tissue for regenerative therapies. This aim is already represented in the first half of the manuscript directly in the *in vitro* experiments with CSPG-containing Matrigel (Figs. 1-3). This analysis under the simple *in vitro* condition (Fig. 1) mimicking the CS-rich injured brain tissue (Fig. 4) showed that growth cone extension and neuronal migration are inhibited by CSPGs. Next, HSPGs were found to attenuate the inhibitory effects of CSPG on growth cones and thus promote neuronal migration (Figs. 2 and 3), leading us to further investigate the effects of HSPGs *in vivo*. Introduction of HSPGs using biomaterial scaffolds resulted in growth cone extension, promotion of neuronal migration, and regeneration (Figs. 5 and 6), leading to functional recovery (Fig. 6). Together, the results of all these experiments (Figs. 1-6) enabled us to conclude that HSPGs are previously unrecognized molecular promoters of growth cone-mediated neuronal migration in CS-rich injured brain and recovery of brain functions.

To clarify the common objective to all experiments conducted in this study, graphical abstracts have been modified (Supplementary Fig. 3k) and added (Supplementary Fig. 8), while the text has been modified as follows:

Introduction

“The binding of one member of this family that is expressed in axonal growth cones, namely $PTP\sigma$, to chondroitin sulfate proteoglycans (CSPGs), extracellular matrix molecules, has been shown to inhibit axon elongation in injured neural tissues¹⁵⁻¹⁸.” (Page 3, lines 12-15)

*“In this study, we investigated the cytoskeletal structure dynamics and molecular functions of GCLs in migrating neurons and found that migrating neurons possess a growth cone that shares important functions with axonal growth cones. In brief, we found that the CSPG/HSPG receptor $PTP\sigma$ is concentrated in migrating-neuron growth cones and regulates growth cone motility and neuronal migration. Functionally, we demonstrated that HSPGs promote neuronal migration induced by growth cone extension in both an *in vitro* system mimicking CSPG-rich injured brain tissue as well as in injured mouse brain tissue with increased CSPGs. Using an artificial HSPG-containing scaffold, we further succeeded in regeneration of mature neurons and functional recovery. Based on these findings, we propose a neuronal migration mechanism wherein growth cones regulate migration through interaction with extracellular environments. Harnessing this mechanism could represent a new strategy for restoring brain functionality post-injury by way of modulating extracellular conditions.”* (Page 3, line 27 to page 4, line 4)

Results

“ $PTP\sigma$ is a CSPG receptor that inhibits axon elongation in CS-rich injured neural tissues, and its inhibition promotes axonal regeneration^{15,38}.” (Page 5, lines 33-34)

“Addition of the HS-degrading enzyme heparinase-III reinstated CSPG-mediated migration of inhibition (Supplementary Fig. 3e f), suggesting that HS indeed enables neuronal migration in the presence of CSPGs in vitro, a condition mimicking that found in injured brain tissue.” (Page 6, lines 5-8)

*“Because the elongation of a tip filopodium in contact with *Sdc2* leads to the extension of an associated growth cone under inhibitory conditions, hereafter we will refer to this filopodium as the leading filopodium (Supplementary Fig. 3k). The significance of leading filopodia was examined in the*

following experiments.” (Page 7, lines 12-16)

“These results suggest that application of HSPG-containing scaffolds to an injured brain can promote neuronal migration, thereby enabling neuronal maturation and facilitating recovery of brain function (Supplementary Fig. 8).” (Page 11, lines 11-14)

Discussion

“The present in vitro and in vivo findings demonstrate that a growth cone exists in migrating neurons and its activities are indispensable for neuronal migration, especially in inhibitory environments. Thus, the findings of this study allows us to propose a new strategy for facilitating neuronal migration in the injured brain.” (Page 11, lines 20-24)

“We showed that, like their axonal counterparts, LP growth cones of migrating neurons express $PTP\sigma$ and collapse in response to CSPGs^{15,17,38}, which are abundant in the injured brain.” (Page 11, lines 29-31)

REVIEWERS' COMMENTS

Reviewer #3 (Remarks to the Author):

I am satisfied with the authors response to my comments.